# Chronic stress impairs autoinhibition in neurons of the locus coeruleus to increase asparagine endopeptidase activity

Hiroki Toyoda[1,2]*[†], Doyun Kim[3][†], Byeong Geon Koh[3], Tomomi Sano[4], Takashi Kanematsu[4], Seog Bae Oh[3,5]*, Youngnam Kang[6]*

[1]Department of Oral Physiology, Graduate School of Dentistry, Osaka University, Osaka, Japan; [2]Department of Physiology, School of Dentistry, Aichi Gakuin University, Nagoya, Japan; [3]Department of Neurobiology & Physiology, School of Dentistry and Dental Research Institute, Seoul National University, Seoul, Republic of Korea; [4]Department of Cell Biology, Aging Science, and Pharmacology, Faculty of Dental Science, Kyushu University, Fukuoka, Japan; [5]ADA Forsyth Institute, Somerville, United States; [6]Department of Behavioral Sciences, Graduate School of Human Sciences, Osaka University, Osaka, Japan

*For correspondence:
toyoda@dpc.agu.ac.jp (HT);
odolbae@snu.ac.kr (SBO);
kang.yn.923@osaka-u.ac.jp (YK)

[†]These authors contributed equally to this work

Competing interest: The authors declare that no competing interests exist.

## eLife Assessment

This **fundamental** study explores a novel cellular mechanism underlying the degeneration of locus coeruleus neurons during chronic restraint stress. The evidence supporting the overexcitation of LC neurons after chronic stress is **compelling**. The topic is timely, the proposed mechanistic pathway is innovative, and the findings have translational relevance, particularly regarding therapeutic strategies targeting α2A-AR internalization in neurodegenerative diseases.

**Abstract** Impairments of locus coeruleus (LC) are implicated in anxiety/depression and Alzheimer's disease (AD). Increases in cytosolic noradrenaline (NA) concentration and monoamine oxidase A (MAO-A) activity initiate the LC impairment through production of NA metabolite, 3,4-dihydroxyphenyl-glycolaldehyde (DOPEGAL), by MAO-A. However, how NA accumulates in soma/dendritic cytosol of LC neurons has never been addressed despite the fact that NA is virtually absent in cytosol while NA is produced exclusively in cytoplasmic vesicles from dopamine by dopamine-β-hydroxylase. Since reuptake of autocrine-released NA following spike activity is the major source of NA accumulation, we investigated whether and how chronic stress can increase the spike activity accompanied by NA autocrine. Overexcitation of LC neurons is normally prevented by the autoinhibition mediated by activation of α2A-adrenergic receptor (AR)-coupled inwardly rectifying potassium-current (GIRK-I) with autocrine-released NA. Patch-clamp study revealed that NA-induced GIRK-I in LC neurons was decreased in chronic restraint stress (RS) mice, while a similar decrease was gradually caused by repeated excitation. Chronic RS caused internalization of α2A-ARs expressed in cell membrane in LC neurons and decreased protein/mRNA levels of α2A-ARs/GIRKs in membrane fraction. Subsequently, chronic RS increased the protein levels of MAO-A, DOPEGAL-induced asparagine endopeptidase (AEP), and tau N368. These results suggest that chronic RS-induced overexcitation due to the internalization of α2A-ARs/GIRK is accompanied by $[Ca^{2+}]_i$ increases, subsequently increasing $Ca^{2+}$-dependent MAO-A activity and NA autocrine. Thus, it is likely that internalization of α2A-AR increased cytosolic NA, as reflected in AEP increases, by facilitating reuptake of autocrine-released NA. The suppression of α2A-AR internalization may have a translational potential for anxiety/AD treatment.

## Introduction

Locus coeruleus (LC) is implicated in anxiety and depression (*Itoi and Sugimoto, 2010*) and is often the first place where Alzheimer's disease (AD)-related pathology appears (*Mather and Harley, 2016*). Noradrenaline (NA) is known to inhibit amyloid-induced oxidative stress and caspase activation in cortex/hippocampus (*Counts and Mufson, 2010*). Thus, the impairment of the LC-noradrenergic system is potentially involved in the AD pathogenesis (*Ross et al., 2015*; *Chen et al., 2022*).

It was reported that cytosolic NA in the soma/dendrites of LC neurons is degraded by mono-amine oxidase A (MAO-A) into 3,4-dihydroxyphenyl-glycolaldehyde (DOPEGAL) (*Burke et al., 1999*), which generates apoptotic free radicals (*Burke et al., 1998*) and activates asparagine endopeptidase (AEP) to produce Aβ and hyperphosphorylated tau, consequently leading to AD (*Burke et al., 1999*; *Kang et al., 2020*). However, there normally appears to be no NA except dopamine in the cytosol of LC neurons because NA is produced exclusively in cytoplasmic vesicles by dopamine-β-hydroxylase (DBH) (*Potter and Axelrod, 1963*) from dopamine which is taken up into cytoplasmic vesicles by vesicular monoamine transporter type 2 (VMAT2).

LC neurons release NA following action potential not only from the axon terminals but also from their cell bodies as autocrine (*Huang et al., 2007*), and autocrine-released NA activates α2A adren-ergic receptor (α2A-AR) expressed on the soma/dendrites (*Aoki et al., 1994*), resulting in activation of α2-AR-coupled inwardly rectifying K$^+$ (GIRK) channel (*Arima et al., 1998*) to cause autoinhibition (spike-afterhyperpolarization) (*Williams et al., 1985*). Thus, overexcitation of LC neurons is normally prevented by the autoinhibition.

As autocrine-released NA is taken up by NA transporter (NAT) (*Torres et al., 2003*), uptaken-NA may be the only source of free NA to be metabolized by MAO-A in the cytosol of LC neurons. However, the amount of free NA to be metabolized by MAO-A in the cytosol is thought to be very small due to a much lower affinity of NA to MAO-A compared to VMAT2 (*Costa et al., 2012*). Therefore, it is not clear how free NA to be metabolized by MAO-A is increased in the soma/dendrites of LC neurons in AD pathogenesis. Because the passive NA leakage occurs from cytoplasmic vesicles in a manner that achieves a dynamic equilibrium with active NA storage into cytoplasmic vesicles by VMAT2 (*Eisen-hofer et al., 2004*), increased NA uptake by NAT in response to increased NA autocrine following $[Ca^{2+}]_i$ increases or overexcitation may increase free NA by facilitating the passive NA leakage from cytoplasmic vesicles. Furthermore, the affinity of MAO-A for NA may be modifiable as MAO-A activity is $Ca^{2+}$-dependent (*Cao et al., 2007*). Then, a question arises whether or not impairment of autoinhi-bition that causes overexcitation and $[Ca^{2+}]_i$ increases can consequently increase AEP activity through the production of DOPEGAL.

Neuronal activity in LC neurons is transiently increased by stress (*Valentino and Foote, 1988*) through activation of corticotropin-releasing factor (CRF) receptor (*Valentino et al., 1983*; *McCall et al., 2015*) together with glutamate receptors (*Valentino et al., 2001*). However, even if firing activity is increased by synaptic activation (*Barcomb et al., 2022*), autoinhibition would occur to suppress overexcitation in LC neurons. Nevertheless, it is known that long-term overexcitation due to chronic stress ultimately causes degeneration of LC neurons (*Kitayama et al., 2004*). Then, this raises an important question of whether and how stress impairs autoinhibition to cause sustained overexci-tation. The whole mechanism for the impairment of LC would be revealed by solving these questions. Here, we examined whether the autoinhibition mediated by α2A-AR-coupled GIRK-I is impaired by overexcitation in LC neurons or chronic stress, to clarify the mechanisms for NA accumulation and/or MAO-A activity increase.

## Results

### Spike frequency adaptation in LC neurons is mediated by α2-AR-coupled GIRK channel

LC neurons invariably display spike-frequency adaptation (*Figure 1A and B*). We have hypothesized that the autoinhibition would occur following respective spike firing and contribute to the generation of the spike-frequency adaptation and the afterhyperpolarization seen at the offset of current pulses.

When the cumulative number of spikes was plotted against the elapsed time from the onset of the current pulse every 50 ms, the cumulative spike number increased nonlinearly in a saturation manner (*Figure 1C*). This nonlinear relationship was fitted with a saturation curve defined by the Monod

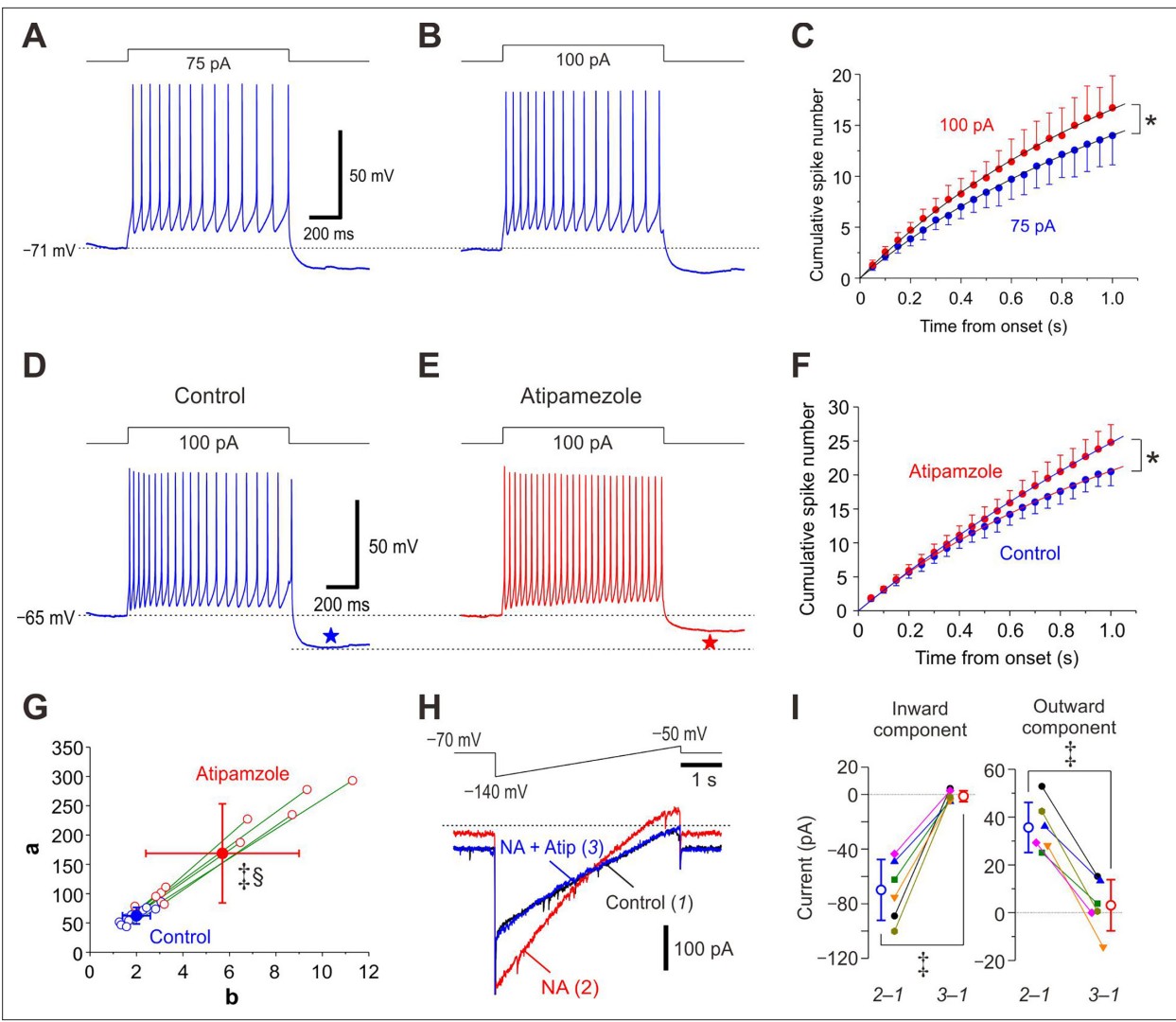

**Figure 1.** Atipamezole abolishes spike-frequency adaptation and blocks cytosolic noradrenaline (NA)-induced GIRK-I in locus coeruleus (LC) neurons. (**A, B**) Sample traces of spike trains induced in an LC neuron by injection of current pulses at 75 and 100 pA. Note the spike-frequency adaptation and pulse-afterhyperpolarization (pulse-AHP). (**C**) Plotting of the cumulative spike numbers vs the elapsed time from the onset of the current pulses every 50 ms at 75 pA (blue circles) or 100 pA (red circles) (*n*=7). Two-way RM ANOVA, *p<0.001. The saturation level (**a**) and the half saturation constant (**b**) were measured by fitting the saturation function defined as *y* = (*a* * *x*)/(*b*+*x*) to the data points. The values of *a* at 75 and 100 pA current pulses were 44.6±16.2 and 52.5±12.3, respectively (paired t-test, p=0.010). The values of *b* at 75 and 100 pA current pulses were 2.1±0.7 and 2.1±0.4, respectively (paired t-test, p=0.823). (**D, E**) Sample traces of spike trains induced in an LC neuron evoked by current pulses at 100 pA under control condition and after application of 10 µM atipamezole. Note that atipamezole suppresses spike-frequency adaptation and pulse-AHP (blue and red stars). (**F**) Plotting of the cumulative spike numbers vs the elapsed time during the current pulses every 50 ms before and after application of atipamezole (blue and red circles, respectively) (*n*=10). Two-way RM ANOVA, *p<0.001. (**G**) Plotting of *a* against *b*, which were measured by fitting the saturation function to the data points in F. Both the values of *a* and *b* obtained by curve fitting to the data points after application of atipamezole (open and filled red circles) were significantly shifted to larger values than those obtained under control condition (open and filled blue circles) (paired t-test, *a* and *b*, ‡p=0.003 and ‡p=0.002, respectively), and there was a significant difference in the relationship between *a* and *b* (Wilk's lambda, §p=0.002) (*n*=10). (**H**) Top, Voltage command pulse. Lower panel, Superimposed current traces obtained under control condition (1), after application of 100 µM NA (2), and after application of 10 µM atipamezole (Atip) in addition to NA (3). The amplitudes of inward and outward components in NA-induced GIRK-I were –70±22 and 36±11 pA, respectively (paired t-test, ‡p=0.005, n=6). (**I**) A graph showing the effects of atipamezole on amplitudes of the inward and outward components of NA-induced GIRK-I at –130 and –60 mV, obtained by subtraction of currents recorded under control condition from those recorded after application of NA (2–1) and from those recorded after application of NA and atipamezole (3–1). Inward component, paired t-test, ‡p<0.001; outward component, paired t-test, ‡p<0.001 (n=6).

The online version of this article includes the following source data and figure supplement(s) for figure 1:

**Source data 1.** Data used for graphs presented in *Figure 1C, F, G, and I*.

**Figure supplement 1.** Spike-frequency adaptation and A-like K+ current in a male juvenile mouse and a female adult mouse.

equation (see Extended Methods) in order to quantify the spike-frequency adaptation. Because there was no significant difference in $b$ between the two groups of spike trains, the profile of the normalized cumulative spike number vs the elapsed time remains the same regardless of firing frequency, suggesting that spike-frequency adaptation occurs in the same way.

The spike-frequency adaptation observed was almost completely abolished by a blocker of α2-AR, atipamezole (*Figure 1D and E*), which subsequently largely decreased pulse-afterhyperpolarization (pulse-AHP) (*Figure 1D and E*, blue and red stars). The cumulative spike number increased nonlinearly in a saturation manner in the control condition while it increased almost in a linear manner after atipamezole application (*Figure 1F*). In the relationship between $a$ and $b$ examined in LC neurons, atipamezole invariably and significantly shifted both $a$ and $b$ in a larger direction, and there was a significant difference in the relationship between $a$ and $b$ (*Figure 1G*).

Bath application of NA induced a GIRK-I in response to the ramp pulse (*Figure 1H*, red trace) whereas addition of atipamezole to NA almost completely abolished the NA-induced GIRK-I (*Figure 1H*, blue trace), showing that GIRK-I reversed at –97 mV. The amplitudes of the inward component of NA-induced GIRK-I were invariably larger than those of the outward component, consistent with the inwardly rectifying property of GIRK-I. Atipamezole significantly decreased the amplitudes of the inward and outward components (*Figure 1I*). Taken together, NA autocrine is likely to be caused following respective spikes in the LC neuron, and NA is accumulated around the LC neuron to activate α2-AR-coupled GIRK channels, to cause spike-frequency adaptation by enhancing AHP.

## Ca²⁺-dependent rundown of NA-induced GIRK-I in LC neurons

In PC12 cells, heterologously expressed α2A-ARs undergo internalization following agonist application (*Taraviras et al., 2002*). Such internalization of α2A-ARs would result in an abolishment of GIRK-I or desensitization given the functional coupling between α2A-AR and GIRK as in LC noradrenergic neurons. However, in LC neurons, NA-induced GIRK-I showed almost no rundown/desensitization after repeating the ramp pulse 20 times in the persistent presence of NA (*Figure 2—figure supplement 1A–D*). Then, we next examined whether GIRK-I shows rundown under the condition of 'increased excitability' or increased $[Ca^{2+}]_i$, because acute/chronic stress increases the excitability of LC neurons (*Borodovitsyna et al., 2018*; *Campos-Lira et al., 2018*) and various channels/receptors are known to be internalized in a Ca²⁺-dependent manner (*Beattie et al., 2000*; *Aziz et al., 2012*; *Zaccor et al., 2020*).

To examine the Ca²⁺ dependency of the rundown of GIRK-I, the respective ramp pulses to evoke GIRK-I were preceded by various pulse trains which can induce Ca²⁺ currents differentially (compare *Figure 2—figure supplement 1F*, *Figure 2B*). After examining various pulse protocols (*Figure 2—figure supplements 1 and 2*), the best protocol to induce a prominent rundown of GIRK-I was as follows (shown in *Figure 2B*). First, bath application of NA induced GIRK-I (*Figure 2A*, red trace; 2) that showed the inwardly rectifying profile as revealed by subtraction of the control response from that obtained after NA application (*Figure 2D*, red trace; 2–1). Subsequently, the combined command pulse (a set of 5 trains of 20 pulses followed by a ramp pulse; *Figure 2B*, upper trace) was applied every minute. The positive pulse trains evoked large inward Ca²⁺ currents in the presence of extracellular 30 mM TEA (*Figure 2B*, lower traces), whereas similar trains of the same positive pulse hardly induced Ca²⁺ currents when applied in the absence of TEA (*Figure 2—figure supplement 1F*, lower trace). With repetition of the combined command pulse 20 times, the NA-induced GIRK-I (*Figure 2C*, red trace; 2) was decreased to almost the same current as the control response (*Figure 2C*, blue trace; 3). Indeed, no GIRK-I remained after 20 times repetition of the combined command pulse as revealed by the subtraction between trace 1 and trace 3 (*Figure 2D*, blue trace; 3– 1). Both the amplitudes of the inward and outward components of the NA-induced GIRK-I promptly decreased to zero along with repetition of the combined command pulse (*Figure 2E*, blue and red circles; *Figure 2F*). Thus, it is likely that NA-induced GIRK-I displays a marked rundown in a Ca²⁺-dependent manner. Furthermore, we confirmed that these rundowns were completely dependent on the number and application timing of the command pulse in the presence or absence of TEA (compare *Figure 2* with *Figure 2—figure supplements 1 and 2*). Depending on the absence or presence of TEA, the single application of a set of 10 trains of 20 pulses differentially (slightly or moderately, respectively) induced the rundown of NA-induced GIRK-I (compare *Figure 2—figure supplements 1E–J and 2A–E*). By contrast, the repetitive application of a single train of 20 positive pulses in the presence of TEA markedly facilitated

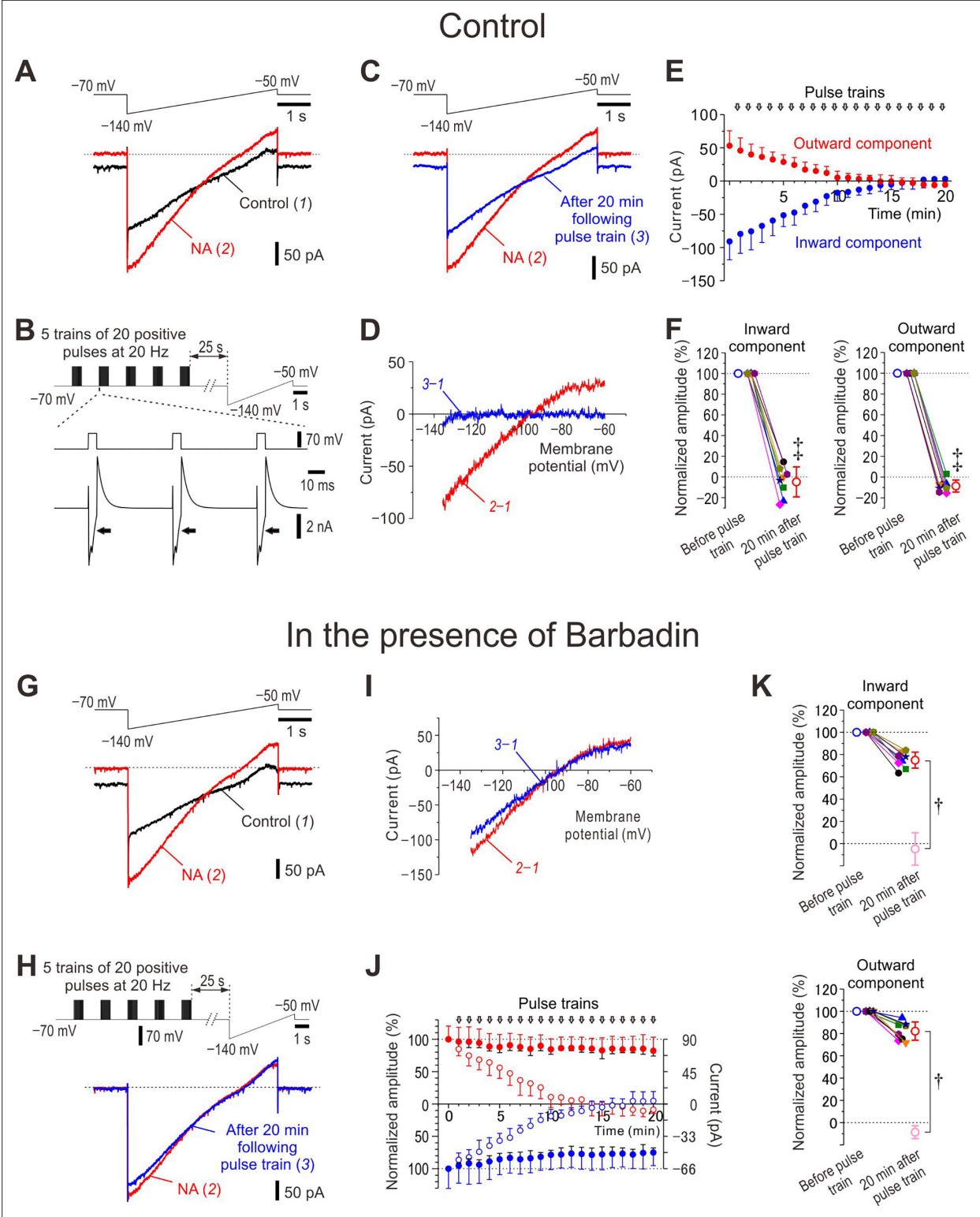

**Figure 2.** Ca$^{2+}$-dependent rundown of noradrenaline (NA)-induced GIRK-I and its suppression by barbadin. (**A**) Upper panel, Ramp command pulse. Lower panel, Superimposed current traces obtained under control condition (1) and after application of 100 µM NA for 5 min (2). (**B**) Upper panel, A combined command pulse applied every minute; five trains of 20 positive pulses (5 ms duration to 0 mV at 20 Hz) at an interval of 2 s in the presence of extracellular 30 mM TEA and intracellular 0.2 mM EGTA, which were followed by the ramp pulse after an interval of 25 s. Lower panel, Representative Ca$^{2+}$ currents in response to positive pulses in the presence of NA and 30 mM TEA. (**C**) Upper panel, Ramp command pulse. Lower panel, Superimposed current traces obtained after application of NA for 5 min (2) and in response to application of the 20th combined pulse in the

*Figure 2 continued on next page*

*Figure 2 continued*

presence of NA and TEA (3). (**D**) The I-V relationship of NA-induced GIRK-I obtained by subtraction of currents recorded under control condition from that recorded after application of NA for 5 min (red trace, 2–1) and that obtained by subtraction of the control current from that recorded in response to application of the 20th combined pulse in the presence of NA and TEA (blue trace, 3–1). (**E**) Plotting of amplitudes of inward components at –130 mV (blue circles) and outward components at –60 mV (red circles) against time. (**F**) Plotting of normalized amplitudes of inward components at –130 mV and outward components at –60 mV before and 20 min after application of positive pulse trains in the presence of NA and TEA. The amplitudes of inward components at –130 mV and those of outward components at –60 mV were normalized by those recorded in the presence of NA before applying positive pulse trains. Inward component, paired t-test, $^{\ddagger}$p<0.001; outward component, paired t-test, $^{\ddagger}$p<0.001 ($n$=8). (**G**) Upper panel, Ramp command pulse. Lower panel, Superimposed current traces obtained under control condition (1) and after application of 100 µM NA for 5 min (2) in LC neurons dialyzed with 100 µM barbadin (β-arrestin/AP2 blocker). (**H**) Upper panel, A combined command pulse applied every minute; five trains of 20 positive pulses (5 ms duration to 0 mV at 20 Hz) at an inter-train interval of 2 s in the presence of extracellular 30 mM TEA and intracellular 0.2 mM EGTA, which were followed by the ramp pulse after an interval of 25 s. Lower panel, Superimposed current traces obtained after application of NA for 5 min (2) and in response to application of the 20th combined pulse in the presence of NA and TEA (3) in LC neurons dialyzed with barbadin. (**I**) The I-V relationship of NA-induced GIRK-I obtained by subtraction of currents recorded under control condition from that recorded after application of NA for 5 min (red trace, 2–1) and that obtained by subtraction of the control current from that recorded in response to application of the 20th combined pulse in the presence of NA and TEA (blue trace, 3–1). (**J**) Plotting of normalized amplitudes (left vertical axis) of inward components at –130 mV (filled blue circles) and outward components at –60 mV (filled red circles) against time ($n$=8). Amplitudes of GIRK-I were normalized to the values obtained before applying pulse trains. Right vertical axis refers to the original amplitudes. Open blue and red circles represent the normalized amplitudes of GIRK-I shown in E. (**K**) Plotting of normalized amplitudes of inward components at –130 mV and outward components at –60 mV before and 20 min after application of positive pulse trains in the presence of NA and TEA ($n$=8). The amplitudes of inward components at –130 mV and those of outward components at –60 mV were normalized to those recorded in the presence of NA before applying pulse trains. Open light pink circles represent results obtained under the control condition in F. Inward component, unpaired t-test, *p<0.001; outward component, unpaired t-test, *p<0.001.

The online version of this article includes the following source data and figure supplement(s) for figure 2:

**Source data 1.** Data used for graphs presented in *Figure 2E, F, J, and K*.

**Figure supplement 1.** Noradrenaline (NA)-induced GIRK currents show no apparent agonist-dependent rundown but show a moderate rundown following application of a train of positive voltage pulses.

**Figure supplement 2.** Differential rundown of noradrenaline (NA)-induced GIRK currents following application of various types of positive voltage pulse trains in the presence of TEA.

**Figure supplement 3.** $Ca^{2+}$ transients in response to current pulse injections in locus coeruleus (LC) neurons.

**Figure supplement 4.** Weakening of spike-frequency adaptation and decrease in pulse-AHP accompany GIRK rundown.

**Figure supplement 5.** Atipamezole blocks rundown of noradrenaline (NA)-induced GIRK currents, waning of spike-frequency adaptation, and suppression of pulse-ADP caused by repetitive application of positive pulse trains.

**Figure supplement 6.** Tertiapin-Q did not block $Ca^{2+}$-dependent rundown of noradrenaline (NA)-induced GIRK currents.

the rundown of NA-induced GIRK-I (*Figure 2—figure supplement 2F–K*). The relationship between the number of spikes (i.e. positive pulse) and $[Ca^{2+}]_i$ was obtained by measuring $Ca^{2+}$ transients using fura-2 (*Figure 2—figure supplement 3*).

To investigate the effects of the rundown of GIRK-I on the spike-frequency adaptation, spike trains were examined before and after the voltage-clamp experiment (*Figure 2A–F*). After the rundown of GIRK-I was observed in the voltage-clamp experiment, the spike-frequency adaptation was absent in the current clamp experiment, and the pulse-ADP was largely reduced (*Figure 2—figure supplement 4A–D*). The cumulative spike number that increased nonlinearly in a saturation manner before rundown of GIRK-I increased almost in a linear manner after rundown of GIRK-I (*Figure 2—figure supplement 4E and F*). Thus, the rundown of GIRK-I invariably resulted in an abolishment of spike-frequency adaptation, similar to the effects of atipamezole on spike-frequency adaptation (*Figure 1*). Therefore, it is likely that $Ca^{2+}$ increases can cause the rundown of NA-induced GIRK-I and almost abolish the spike-frequency adaptation, leading to an increase in the excitability of LC neurons. These observations suggest that $[Ca^{2+}]_i$ increases cause rundown of NA-induced GIRK-I to suppress auto-inhibition, leading to excitability increases that cause further increases in $[Ca^{2+}]_i$ to further cause a rundown of the GIRK-I.

It was also examined whether the rundown of NA-induced GIRK-I can be caused solely by $[Ca^{2+}]_i$ increases without activation of α2-AR. We found that NA-induced GIRK-I subjected to the combined command pulse 20 times in the presence of atipamezole was significantly restored after washout of atipamezole in the presence of NA (*Figure 2—figure supplement 5A–D*). This observation indicates that GIRK-I was not downregulated solely by $[Ca^{2+}]_i$ increases without activation of α2-AR, clearly

revealing that both the activation of α2-AR and $[Ca^{2+}]_i$ increases are necessary to cause the downregulation of GIRK-I. In agreement with this result, the spike-frequency adaptation reflecting the auto-inhibition was also preserved in spite of $[Ca^{2+}]_i$ increases as long as activation of α2-AR is prevented (*Figure 2—figure supplement 5E–J*). We further examined whether the rundown of NA-induced GIRK-I can be prevented by the presence of a GIRK channel blocker, tertiapin-Q. GIRK channel blockade did not prevent the rundown of NA-induced GIRK currents following $[Ca^{2+}]_i$ increases (*Figure 2—figure supplement 6A–H*). In agreement with this result, the spike-frequency adaptation was not preserved by the blockade of GIRK channels during $[Ca^{2+}]_i$ increases (*Figure 2—figure supplement 6I–N*). These findings showed that α2-AR blockade but not GIRK blockade prevented the rundown of NA-induced GIRK-I following $[Ca^{2+}]_i$ increases. Taken together, it is suggested that the rundown of NA-induced GIRK-I may be brought about by the internalization of α2-AR in a $Ca^{2+}$-dependent manner, regardless of opening or closing of GIRK channels.

## β-Arrestin is involved in the rundown of NA-induced GIRK-I

We further examined whether the internalization of α2-AR is involved in the rundown of NA-induced GIRK-I or not. As the β-arrestin/AP2 complex is known to be involved in the agonist-dependent internalization of GPCR (*DeGraff et al., 1999*; *Cottingham et al., 2011*), we examined the effects of barbadin, which prevents AP2 from binding with β-arrestin by forming barbadin/AP2 complex, on the rundown of NA-induced GIRK-I. With the patch pipettes filled with barbadin-including internal solution, it was examined whether NA-induced GIRK-I displays rundown in response to the same combined command pulse (*Figure 2H*, upper trace) as that in the control experiment (*Figure 2B*, upper trace). The NA-induced GIRK-I (*Figure 2G–I*, red traces) was hardly decreased (*Figure 2H and I*, blue traces) after repeating the combined pulse 20 times, in contrast to the control rundown (*Figure 2C and D*). Barbadin largely suppressed the rundown of both components of GIRK-I (*Figure 2J and K*). These results suggest that the rundown of NA-induced GIRK-I is caused by the internalization of α2-AR-coupled GIRK channels.

## RS alters α2A-AR expression in LC neurons

We next investigated whether the expression of α2A-AR is altered in LC neurons by applying restraint stress (RS) to mice. After mice were subjected to daily 12 hr RS for 3 days followed by non-stress condition for 2 days, the immunoreactivity for α2A-ARs (see *Figure 3—figure supplement 1*) seen along the membrane of TH-positive LC neurons was apparently lower compared to the control (*Figure 3A and B*). Consistent with this observation, western blot analysis revealed significantly lower protein levels of α2A-ARs in LC neurons obtained after 3-day RS compared to the control (*Figure 3C*). Especially when α2A-AR expression was examined in the membrane fraction identified using $Na^+/K^+$-ATPase, the immunoreactivity for α2A-AR expression was apparently lower in LC neurons of RS mice than in those of the control mice (*Figure 3D and E*; *, compare the rightmost lower panels). Consistent with this observation, western blot analysis revealed a significantly lower protein level of α2A-ARs in the membrane fraction of LC neurons of RS mice compared to the control, whereas there was no significant difference in the cytosol between the control and RS mice (*Figure 3F and G*). Thus, the internalization of α2A-ARs is not necessarily accompanied by increases in cytosolic levels of α2A-ARs because the internalized receptors are either recycled or degraded (*Koenig and Ikeda, 1996*; *Takei et al., 1996*).

## RS reduces NA-induced GIRK-I

As 3-day RS caused the internalization or downregulation of α2A-ARs, we next examined if the daily 12 hr RS for up to 5 days can differentially cause reductions of NA-induced GIRK-I in LC neurons. NA-induced GIRK-I decreased with an increase in the RS period (*Figure 4A–E*). The amplitudes of inward and outward components of GIRK-I decreased with the increase in the period of RS (*Figure 4F and G*), in a way that is described by a saturation function (red interrupted lines). The amplitudes of inward and outward components of GIRK-I in 1-day mice were significantly smaller than those of the control mice, and those in 3-day RS mice were significantly smaller than those in 1-day RS mice, while there was no significant difference in those between 3-day and 5-day RS mice (*Figure 4F and G*). Thus, the downregulation of GIRK-I was increased as the RS period was increased, but in a saturation manner.

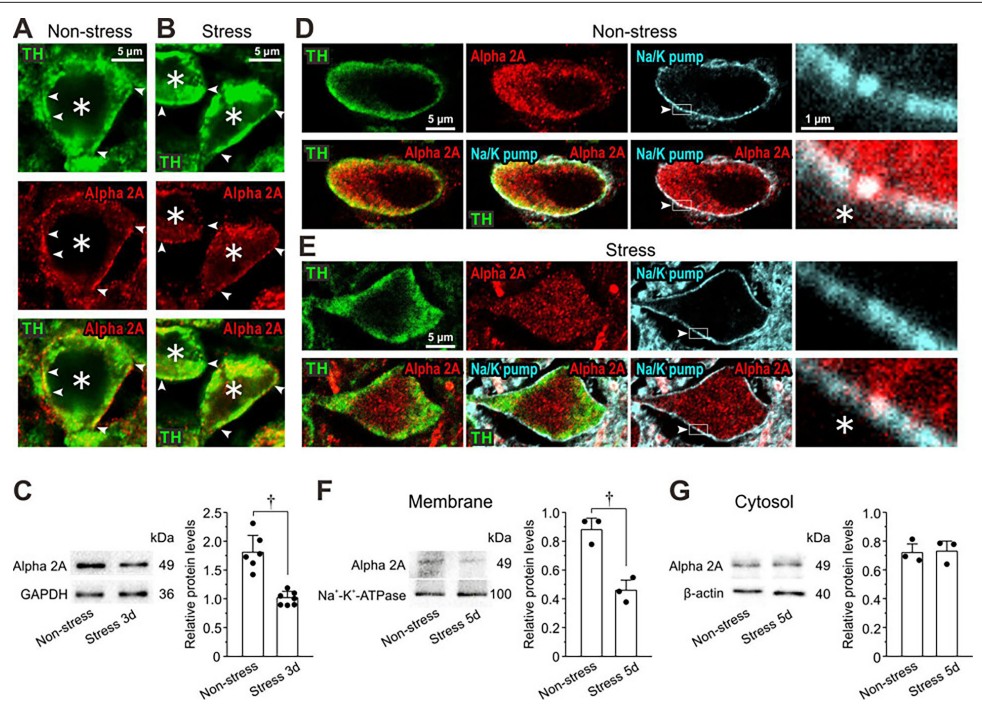

**Figure 3.** Restraint stress (RS) reduces expression of α2A-ARs in locus coeruleus (LC) neurons. (**A, B**) Confocal images of LC neurons showing immunoreactivities for TH and α2A-ARs, together with a merged one in non-stress and 3-day RS mice. Arrowheads indicate the membrane regions of TH-positive neurons, along which α2A-ARs were differentially expressed between the control and the RS mice. Asterisks indicate LC neurons. (**C**) Western blotting analyses showing the expression of α2A-ARs in non-stress and 3-day RS mice (*n*=6 and 7, respectively). Unpaired t-test, †p<0.001. (**D, E**) Confocal images of LC neurons in non-stress and RS mice. Upper panels from left to right showing the respective immunoreactivity for TH, α2A-ARs (with rabbit anti-α2A-AR) and Na$^+$-K$^+$-pump, and an enlarged image of the region enclosed with a rectangle (arrowheads) in its immediate left panel. Lower panels from left to right showing the merged image of TH and α2A-ARs, that of TH, α2A-ARs, and Na$^+$-K$^+$-pump, that of α2A-ARs and Na$^+$-K$^+$-pump, and an enlarged image of the region enclosed with a rectangle in its immediate left panel. (**F, G**) Western blotting analyses showing expressions of α2A-ARs and Na$^+$-K$^+$-pump in membrane fraction and those of α2A-ARs and β-actin in cytosol fraction in non-stress and 5-day RS mice (*n*=3 and 3 samples, respectively). Each sample represents the analysis result in the LC tissues obtained from two to three mice. Membrane; unpaired t-test, †p=0.016. Cytosol; unpaired t-test, †p=0.083.

The online version of this article includes the following source data and figure supplement(s) for figure 3:

**Source data 1.** Original files for the western blot analysis in *Figure 3C, F, and G*.

**Source data 2.** PDF containing the original blots in *Figure 3C, F, and G* with the relevant bands clearly labeled.

**Source data 3.** Data used for graphs presented in *Figure 3C, F, and G*.

**Figure supplement 1.** Reliability of anti-α2A-AR antibody revealed by differential expression of α2A ARs between locus coeruleus (LC) and MTN.

Quantitative PCR analysis revealed that LC neurons expressed much higher mRNA level of α2A-AR than that of α2C-AR (*Figure 4H*) and also expressed higher mRNA level of GIRK1 compared to GIRK2 while GIRK3 was hardly expressed (*Figure 4I*). Consistent with the significant decreases in NA-induced GIRK-I in the mice subjected to 3-day RS, mRNA levels of α2A-AR, GIRK1, and GIRK2 in mice subjected to 3-day RS were significantly smaller than those of the control mice (*Figure 4J–L*). Thus, RS downregulated the expression/production of α2A-AR-coupled GIRK1 prominently in LC neurons.

## Histological and biochemical changes in LC neurons after 5-day RS

Next, immunohistochemical experiments demonstrated that 5-day RS appeared to slightly increase the expressions of TH and MAO-A in LC neurons (*Figure 5—figure supplement 1A*). Furthermore, AEP expression also appeared to increase in cytosol after 5-day RS compared to the control

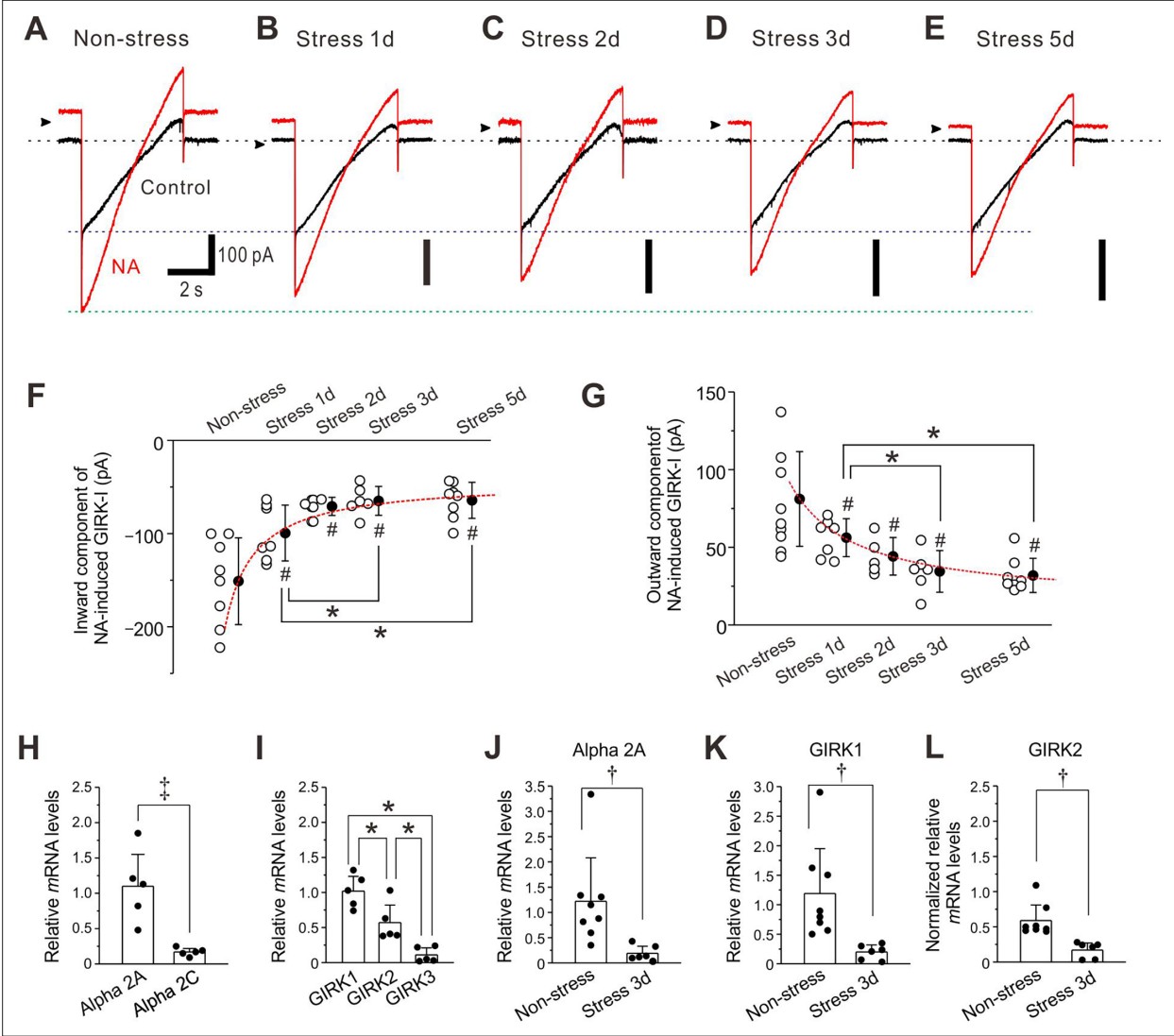

**Figure 4.** Restraint stress (RS) reduces noradrenaline (NA)-induced GIRK-I in locus coeruleus (LC) neurons. (**A–E**) Representative traces of NA-induced GIRK-I obtained from LC neurons in non-stress and 1-day, 2 day, 3-day, and 5-day RS mice. (**F, G**) Amplitudes of inward and outward components of NA-induced GIRK-I obtained from LC neurons in non-stress and 1-day, 2-day, 3-day, and 5-day RS mice (n=8, 7, 5, 6, and 8, respectively) decreased with the increase in the period of RS, in a way that can be described by a saturation function (red interrupted lines). The saturation level (a+b) and the half saturation constant (c) were determined by fitting the saturation function, defined as $y=a + (b* x)/(c+x)$, to the data points. The values of a, b, and c for the inward component of GIRK-I were –151.1, 108, and 0.9, respectively, and those for the outward component of GIRK-I were 81.1, –67, and 1.6, respectively. Inward component: one-way ANOVA, $p<0.001$, post hoc Fisher's PLSD, 1 day; #$p<0.001$ vs non-stress and *$p <0.05$ vs 3 day and 5 day, 2 day; #$p<0.001$ vs non-stress, 3 day; #$p<0.001$ vs non-stress, 5 day; #$p<0.001$ vs non-stress. Outward component: one-way ANOVA, $p<0.001$, post hoc Fisher's PLSD, 1 day; $p=0.004$ vs non-stress and *$p<0.05$ vs 3 day and 5 day, 2 day; $p=0.015$ vs non-stress, 3 day; $p<0.001$ vs non-stress, 5 day; $p<0.001$ vs non-stress. (**H**) Relative expressions of α2A and α2C mRNAs, normalized to GAPDH in LC neurons (n=5). Paired t-test, ‡$p=0.014$. (**I**) Relative expressions of GIRK1, GIRK2, and GIRK3 mRNAs, normalized to GAPDH in LC neurons (n=5). One-way RM ANOVA, $p<0.001$, post hoc Fisher's PLSD; *$p=0.008$ for GIRK1 vs GIRK2, *$p<0.001$ for GIRK1 vs GIRK3, *$p=0.008$ for GIRK2 vs GIRK3. (**J, K**) Relative expressions of α2A and GIRK1 mRNAs, respectively, normalized to GAPDH in LC neurons in non-stress mice (n=8) and 3-day RS mice (n=6). α2A-AR: unpaired t-test, †$p=0.020$; GIRK1: unpaired t-test, †$p=0.013$. (**L**) Normalized relative expressions of GIRK2 mRNA in LC neurons in non-stress mice (n=8) and 3-day RS mice (n=6), normalized to the ratio of the mean value of the relative expressions of GIRK1 mRNA to that of GIRK2 mRNA in LC neurons in non-stress mice (**I**). Unpaired t-test, †$p=0.037$.

The online version of this article includes the following source data for figure 4:

**Source data 1.** Data used for graphs presented in *Figure 4F–L*.

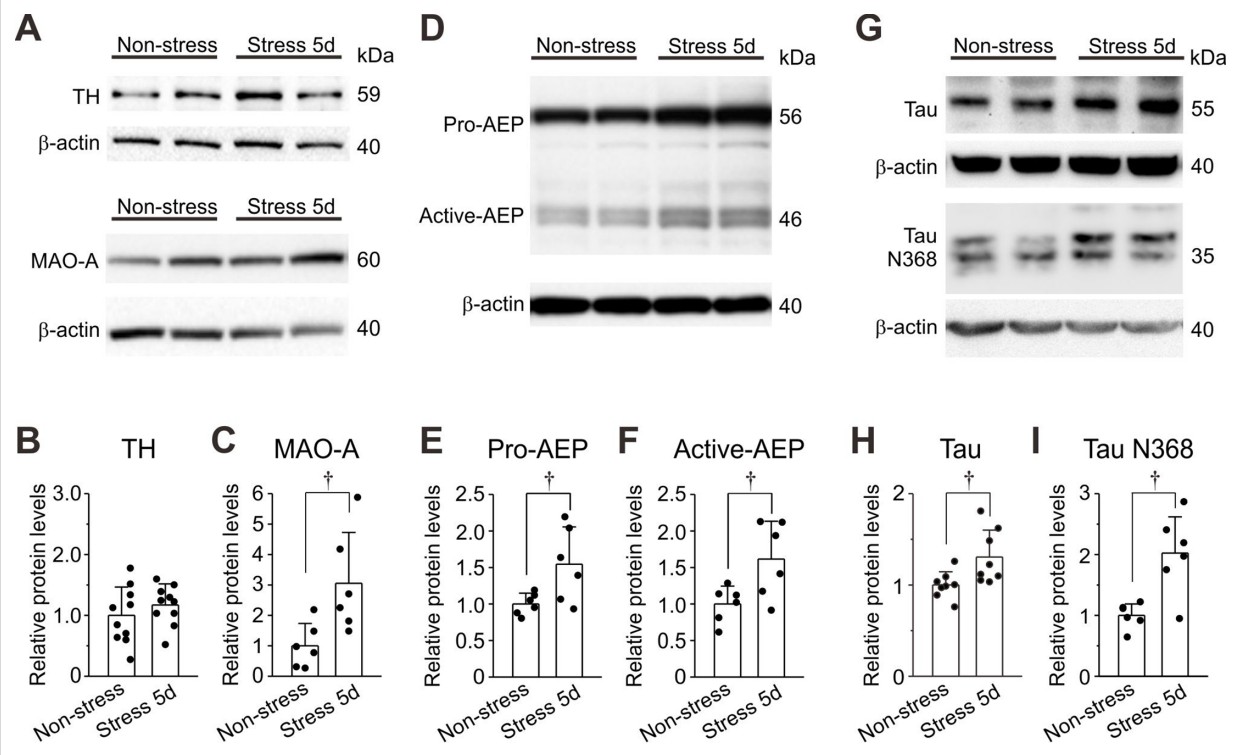

**Figure 5.** Expressions of TH, monoamine oxidase A (MAO-A), dopamine-β-hydroxylase (DBH), asparagine endopeptidase (AEP), and tau proteins in locus coeruleus (LC) neurons. (**A**) Western blotting of TH and MAO-A in LC neurons obtained from non-stress and 5-day RS mice. (**B, C**) Western blot analysis revealing no significant change in TH expression (*n*=10 samples) but a significant increase in MAO-A expression (*n*=6 samples) in 5-day RS group compared to non-stress group. Unpaired t-test, †p=0.029. (**D**) Western blotting of pro-AEP and active AEP in LC neurons obtained from non-stress and 5-day RS mice. (**E, F**) Western blot analysis revealing significant increases in pro-AEP and active AEP (*n*=6 and *n*=6 samples, respectively) in 5-day RS group compared to non-stress group. Unpaired t-test, †p=0.034 (**E**) and †p=0.047 (**F**). (**G**) Western blotting of tau and tau N368 fragment in LC neurons obtained from non-stress and 5-day RS mice. Unpaired t-test, †p=0.025. (**H, I**) Western blot analyses revealing a significant increase in tau and tau N368 fragment in LC neurons (*n*=8 and *n*=6, respectively) in 5-day RS group compared to non-stress group. Each sample represents the analysis result in the LC tissues obtained from two to three mice. Unpaired t-test, †p<0.05.

The online version of this article includes the following source data and figure supplement(s) for figure 5:

**Source data 1.** Original files for the western blot analysis in *Figure 5A, D, and G*.

**Source data 2.** PDF containing the original blots in *Figure 5A, D, and G* with the relevant bands clearly labeled.

**Source data 3.** Data used for graphs presented in *Figure 5B, C, E, F, H, and I*.

**Figure supplement 1.** Immunoreactivities of TH, monoamine oxidase A (MAO-A), dopamine-β-hydroxylase (DBH), and asparagine endopeptidase (AEP) in locus coeruleus (LC) neurons.

(*Figure 5—figure supplement 1B*). Because TH catalyzes the rate-limiting step in the synthesis of catecholamines, if the RS enhances TH activity, free NA concentration in LC neurons would be directly and greatly increased. Therefore, we first carefully examined whether TH is really upregulated in RS mice or not by performing western blot analysis. No significant increase in the protein level of TH was observed in 5-day RS mice compared to the control (*Figure 5A*, upper panel; *Figure 5B*). This result clearly indicates that TH activity was not significantly upregulated in 5-day RS mice, in spite of the TH immunohistochemical experiment. We further examined whether protein levels of MAO-A and AEP are increased or not by 5-day RS. Western blot analysis revealed significant increases in the protein levels of MAO-A (*Figure 5A*, lower panel; *Figure 5C*) and of pro-AEP and active AEP (*Figure 5D–F*) in LC neurons after 5-day RS compared to the control. Western blot analysis also revealed a significant increase in the expression of tau and tau N368 fragment after 5-day RS (*Figure 5G–I*). Thus, an increased activity of active AEP caused a production of tau N368 fragments as observed at 37 and 35 kDa (*Figure 5G*), which is consistent with a previous report (*Zhang et al., 2014*). Relative protein levels of the tau N368 fragments (*Figure 5I*) were the mean values obtained from the two band

intensities, which may reflect the two N-terminal fragments (upper and lower bands) generated by AEP cleavage at Asn-368 of the 4 and 3 repeat-tau isoforms, respectively (*Goedert et al., 1989*). Finally, the Y-maze test revealed that 5-day RS decreased spatial memory (*Figure 6—figure supplement 1*).

## Discussion

We have found that LC neurons display spike-frequency adaptation, which is caused by the autoinhibition mediated by α2A-AR-coupled GIRK-I following respective spikes (*Figure 1*) and that α2A-AR-coupled GIRK-I showed $Ca^{2+}$-dependent rundown due to the internalization of α2A-AR (*Figure 2*, *Figure 2—figure supplement 5A–D*, and *Figure 2—figure supplement 6E–H*), leading to the abolishment of spike-frequency adaptation and the excitability increase (*Figure 2—figure supplement 4*). Immunohistochemical examination and western blot analysis revealed that α2A-AR expression in cell membrane was decreased following RS (*Figures 3 and 4J*). Consistently, NA-induced GIRK-I was decreased in LC neurons with an increase in the period of RS (*Figure 4A–G*). Following chronic RS, protein levels of MAO-A, pro-/active-AEP, and tau and tau N368 fragment were significantly increased (*Figure 5A, C–I*).

Based on these results, we propose a novel mechanism for how chronic RS impairs autoinhibition and increases cytosolic free NA to be metabolized by enhanced $Ca^{2+}$-dependent MAO-A activity, to consequently increase AEP activity and produce tau N368 protein (*Figure 6*). In non-stress mice, autocrine-released NA is slowly taken up by NAT after binding to α2A-ARs to cause autoinhibition (*Figure 6A*), while in chronic RS mice, autocrine-released NA may be directly and rapidly taken up by NAT due to the absence (internalization) of α2A-ARs (*Figure 6B*). Such an increased rate of reuptake of NA by NAT would increase free NA concentration by facilitating the leakage of NA from cytoplasmic vesicles in the cell body, which would be metabolized by upregulated MAO-A activity to DOPEGAL, as revealed by the increased activity of AEP in chronic RS mice (*Figure 6B*). It may be further necessary to perform an enzymatic assay to better demonstrate the AEP and MAO-A activities.

### Spike-frequency adaptation

LC neurons display spike-frequency adaptation. The mechanism for the spike-frequency adaptation may be different among neurons while M currents are usually responsible for the adaptation in hippocampal pyramidal neurons (*Aiken et al., 1995*). In LC neurons, the spike-frequency adaptation was mediated by α2A-AR-coupled GIRK-I as the spike-frequency adaptation was abolished by atipamezole (*Figure 1D–G*) and by causing the rundown of GIRK-I (*Figure 2—figure supplement 4*). These results suggest that autocrine NA release following respective spikes activated α2A-ARs expressed in the soma/dendrites of LC neurons and subsequently activated α2A-AR-coupled GIRK-I to cause autoinhibition as spike-frequency adaptation. Thus, spike-frequency adaptation clearly reflects NA autocrine.

### Rundown of α2A-AR-coupled GIRK-I

α2A-AR-coupled GIRK-I displayed $Ca^{2+}$-dependent rundown in the presence of agonist. The degree and time course of rundown of α2A-AR-coupled GIRK-I were dependent on the number and timing of positive pulses applied under voltage-clamp condition to cause $Ca^{2+}$ currents (*Figure 2A–F*, *Figure 2—figure supplement 1E–J*, *Figure 2—figure supplements 2 and 3*). Given the $Ca^{2+}$ dependency of the rundown of autoinhibition, the excitability of LC neurons would increase as LC neurons were excited more such as under RS condition, resulting in a vicious cycle between excitability increase accompanied by $[Ca^{2+}]_i$ increase and rundown of autoinhibition/$Ca^{2+}$-dependent internalization of α2A-ARs.

### Internalization of α2A-ARs

Activation of GPCR by agonist binding usually causes the formation of GPCR/β-arrestin complexes. Subsequently, the GPCR/β-arrestin complex is assembled through the interaction between β-arrestin and AP2 in clathrin-coated pits (CCPs), which is formed by the association of AP2 with phosphatidylinositol 4,5-bisphosphate (PIP2) at the plasma membrane, leading to endocytosis of GPCRs (*Pierce and Lefkowitz, 2001*). PIP2 is locally produced from the membrane phosphatidylinositol (PI) by PI4/PIP5 kinases in a $Ca^{2+}$-dependent manner (*Sarmento et al., 2014*). Barbadin prevented the stable formation of sufficient GPCR/β-arrestin/AP2 complexes in CCPs as barbadin prevents AP2 from binding

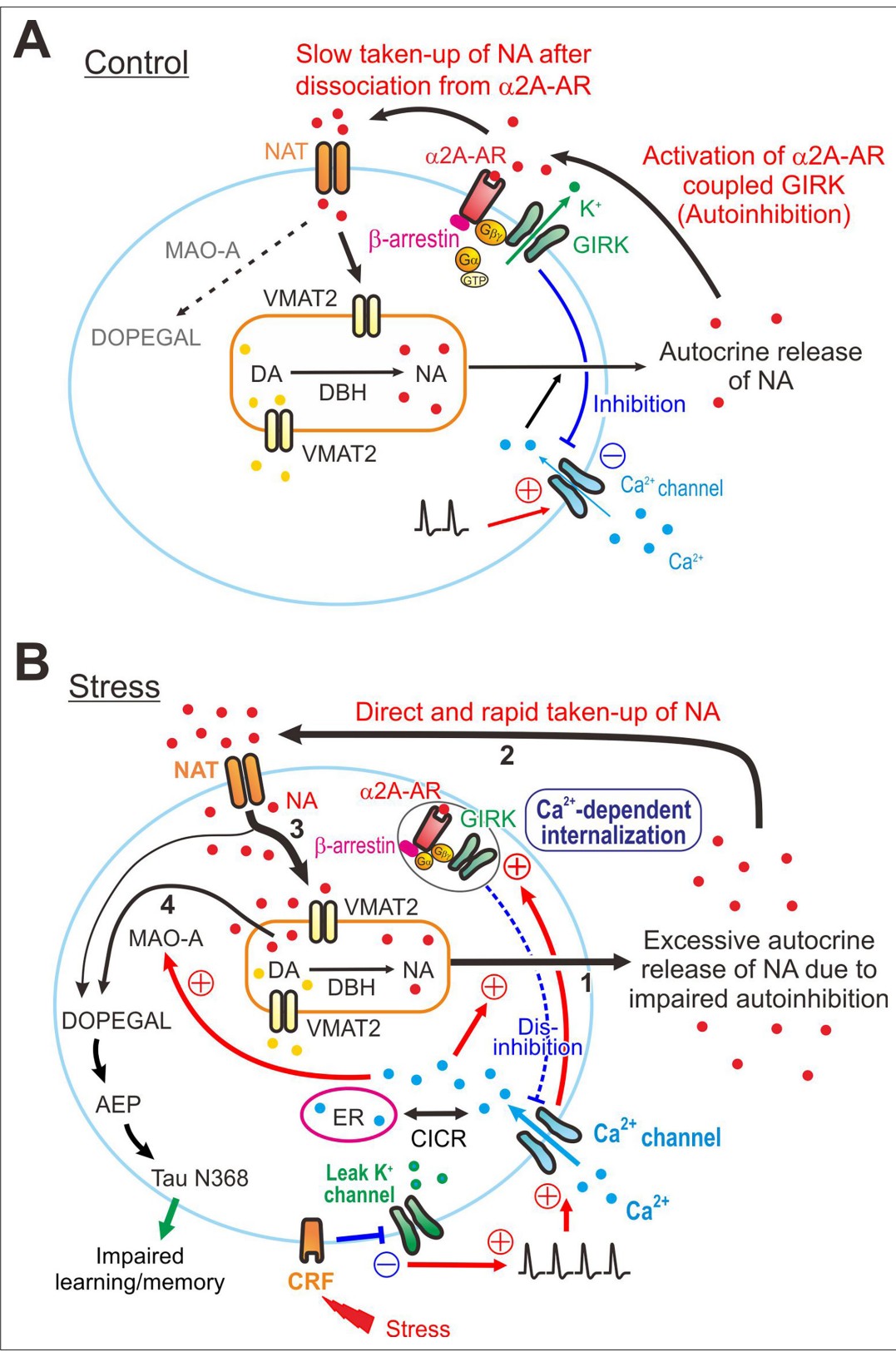

**Figure 6.** Presumed cellular mechanisms for the degeneration of locus coeruleus (LC) by stress. (**A, B**) Differential free concentrations of noradrenaline (NA) to be metabolized by monoamine oxidase A (MAO-A) into 3,4-dihydroxyphenyl-glycolaldehyde (DOPEGAL) between control and stress conditions: Under the control condition (**A**), NA in the cytosol is mostly taken up into the cytoplasmic vesicles by VMAT2 (thick arrow 3), rather

*Figure 6 continued on next page*

*Figure 6 continued*

than being directly metabolized by MAO-A into DOPEGAL (thin interrupted arrow). Vesicular NA is released from cell bodies as autocrine following $[Ca^{2+}]_i$ increases caused by action potentials, and the released NA activates α2A-AR-coupled GIRK channels, causing autoinhibition. Subsequently, the autocrine-released NA is slowly taken up into the cytosol of LC neurons by NAT after dissociation from α2A-ARs. Under the stress condition (**B**), an activation of CRF receptors in LC neurons by stress inhibits leak $K^+$ channels and increases firing activities in LC neurons, subsequently causing a larger $[Ca^{2+}]_i$ increase together with $Ca^{2+}$-induced $Ca^{2+}$ release (CICR). Impairment of autoinhibition due to $Ca^{2+}$-dependent internalization of α2A-ARs-coupled GIRK channels leads to the persistent excitation in LC neurons, which enhances autocrine release of NA (thick arrow 1). Subsequently, the excessively autocrine-released NA is taken up directly and rapidly by NAT into the cytosol without binding to α2A-ARs (thick arrow 2). Such a facilitation of reuptake of NA by NAT would increase active NA storage into cytoplasmic vesicles by VMAT2 (thick arrow 3), while the rate of NA leakage from cytoplasmic vesicles would also increase (arrow 4) due to a dynamic equilibrium in cytoplasmic vesicles between active NA storage into cytoplasmic vesicles and passive NA leakage from cytoplasmic vesicles. Subsequently, such an increase in the rate of NA leakage would result in an increase in an MAO-A metabolite, DOPEGAL, and AEP, leading to a production of cleaved tau N368 fragment and an impairment of learning/memory.

The online version of this article includes the following figure supplement(s) for figure 6:

**Figure supplement 1.** Restraint stress (RS)-induced impairment of spatial memory.

with β-arrestin by forming barbadin/AP2 complex, thereby hampering GPCR internalization (*Beautrait et al., 2017*). In the present study, intracellular barbadin effectively suppressed the rundown of GIRK-I (*Figure 2G–K*), indicating that the rundown of GIRK-I was caused by the internalization of α2A-ARs in LC neurons.

## Chronic RS increases the activities of MAO-A and pro-/active-AEP in the cell body

Chronic cold stress reduced α2-AR-mediated inhibition of LC neurons (*Jedema et al., 2008*), and chronic social stress downregulated α2-ARs in the LC (*Flügge, 1996*). Our present results were consistent with those previous findings: Following RS, α2A-AR expression was decreased due to the internalization of α2A-ARs (*Figure 3*), together with the decreases in NA-induced GIRK-I (*Figure 4A–G*) which mediates the autoinhibition. As long as RS persists, the impairment of autoinhibition leads to the persistent overexcitation in LC neurons. This would further increase NA autocrine, because respective spikes were accompanied by NA-autocrine as reflected in spike-frequency adaptation or autoinhibition.

As extracellular NA is mostly taken up into cytosol by NAT (*Torres et al., 2003*), it is likely that under chronic RS, the increased autocrine-released NA is directly and more rapidly taken up by NAT due to the absence of α2A-ARs (*Figure 6B*). Such an increase in the rate of NA uptake by NAT is similar to the effects of uptake inhibitor of transmitter ligand on the activity of receptors. After reuptake of NA by NAT, a large portion of NA in the cytosol is restored in cytoplasmic vesicles by VMAT2, while only a small portion of NA can escape from VMAT2 to be metabolized by MAO-A (*Eisenhofer et al., 1992*), because VMAT2 has a much higher affinity for NA than MAO-A (*Bloom, 2006*; *Costa et al., 2012*). However, there is an additional component of NA to be metabolized by MAO-A, as NA can leak from cytoplasmic vesicles into cytosol (*Eisenhofer et al., 2004*). When reuptake of NA by NAT is facilitated due to the internalization of α2A-ARs in response to NA autocrine, active NA storage into cytoplasmic vesicles by VMAT2 would increase. Then, the rate of NA leakage from cytoplasmic vesicles would also increase because passive NA leakage from cytoplasmic vesicles and active NA storage into cytoplasmic vesicles by VMAT2 are in a dynamic equilibrium in cytoplasmic vesicles (*Eisenhofer et al., 2004*). Subsequently, such increases in the rate of NA leakage would result in increases in a MAO-A metabolite, DOPEGAL (*Goldstein, 2021*; *Figure 6B*). Indeed, it has previously been demonstrated that a partial inhibition of NAT in human subjects reduced the original level of a derivative of DOPEGAL (3,4-dihydroxyphenylglycol) in CSF by more than 50% (*Kielbasa et al., 2015*). Thus, free NA to be metabolized by MAO-A would be increased when autocrine-released NA is taken up by NAT. On the other hand, it is also known that the decreased VMAT2 levels, generated in VMAT2 heterozygote mice, increased the vulnerability of nigral dopamine neurons to MPTP (*Takahashi et al.,*

*1997*). Thus, it is conceivable that NA leakage for free NA accumulation is influenced by both NAT and VMAT2.

Therefore, under the chronic RS condition, more NA leaking from cytoplasmic vesicles is likely to be metabolized by MAO-A, whose activity is enhanced in a $Ca^{2+}$-dependent manner, and by chronic RS (*Cao et al., 2007*; *Figures 5A, C , and 6B*), to DOPEGAL compared to the control condition, as revealed by the increased protein levels of pro-AEP and active-AEP (*Figure 5—figure supplement 1B* and *Figure 5D–F*). Thus, reuptaken NA by NAT may largely be degraded to DOPEGAL by the increased activity of MAO-A. Subsequently, DOPEGAL converts pro-AEP into active-AEP to produce tau N368 protein (*Figure 5G–I*).

It has never been addressed how DOPEGAL production was caused in the cell body of LC neurons while the DOPEGAL production in the cell bodies of LC neurons is well established in AD patients (*Burke et al., 1999*; *Kang et al., 2020*) and in transgenic AD model mice (*Kang et al., 2020*). We demonstrated for the first time that 5-day RS can impair the autoinhibition by causing the internalization of α2A-AR and consequently induce active-AEP to produce tau N368 fragment. Indeed, 5-day RS induced spatial memory impairment-like responses in mice when assessed using Y-maze test (*Figure 6—figure supplement 1*) while anxiety-like behavior was also induced in mice after 3-day RS when assessed using an elevated plus maze test (figure not shown), both suggesting an impairment of LC neurons. It is likely that, in addition to NAT inhibitor, the suppression of the internalization of α2A-AR has a strong translational potential for developing drugs that prevent the degeneration of LC.

## Limitations of this study

One may argue against the usage of 3-week-old juvenile mice to study the neurodegenerative disorder which is normally found in aged ones. However, a stringent autoinhibition together with potent NAT activity was previously demonstrated in adult rats using intracellular recordings (*Williams et al., 1985*). Thus, regardless of the age of mice, LC neurons showed the autoinhibition that is mediated by the activity of α2A-AR-coupled GIRK channels. We also confirmed that spike-frequency adaptation reflecting autoinhibition can be induced in adult mice as prominently as in juvenile mice regardless of sexes (*Figure 1—figure supplement 1*), although we did not investigate the sex differences in RS effects (*Leistner and Menke, 2020*). The downregulation of autoinhibition by the internalization of α2A-AR is agonist- and $Ca^{2+}$-dependent regardless of the age of mice. However, it is not known if there is any difference in the susceptibility of α2A-AR internalization to RS between juvenile and aged mice. Thus, a further study may be necessary to examine if autoinhibition in LC neurons is more easily downregulated in aged mice compared to juvenile ones.

It has also been reported that reuptake of NA by NAT in LC neurons may decrease with age (*Sanders et al., 2005*; *Mitchell et al., 2017*). However, NAT expression in LC in 3-week-old juvenile mice was only slightly larger (by 4%) than in adult mice, although the difference was significant (*Mitchell et al., 2017*). Potent NAT activity stringently affecting autoinhibition was reported in adult rats (*Williams et al., 1985*). Furthermore, partial inhibition of NAT in human subjects largely reduced the original level of a derivative of DOPEGAL in CSF as described above. Thus, based on the present study, it is likely that increased reuptake of NA by NAT following excessive NA autocrine in the absence of α2A-AR, induced in response to chronic RS, resulted in the increase in active-AEP and subsequent production of tau N368. However, a further experiment to directly assess NAT activity would be necessary to firmly justify our hypothesis.

To further validate the mechanisms which we have proposed in the present study (*Figure 6*), direct quantification of the relationship between α2A-AR internalization and increased cytosolic NA levels may be desirable. However, at present, the quantification of such connection may not be possible because there appears to be no fluorescence probe to label cytosolic NA. None of the NA fluorescence probes, such as NS521, BPS3, or GRAB-NE, is applicable to detect cytosolic free NA because NS521 and BPS3 anchor on plasma membrane and GRAB-NE is dependent on GPCR activation (*Hettie and Glass, 2014*; *Feng et al., 2019*; *Mao et al., 2023*).

## Conclusion

Our study unveiled the cellular mechanism for the degeneration of LC neurons under chronic RS as follows: Chronic RS impaired the autoinhibition in LC neurons by inducing $Ca^{2+}$- and agonist-dependent internalization of soma/dendritic α2A-AR coupled with GIRK channels. This subsequently caused overexcitation/$[Ca^{2+}]_i$ increases and increased $Ca^{2+}$-dependent MAO-A activity and active AEP activity, leading to a production of tau N368 fragment together with an impairment of spatial memory. These results clearly indicate that chronic RS increases cytosolic free NA to be metabolized by MAO-A, presumably by facilitating the reuptake of autocrine-released NA due to the absence/internalization of α2A-AR. Thus, in addition to NAT inhibitor, the suppression of α2A-AR internalization may have a strong translational potential for developing drugs that prevent the degeneration of LC. However, the present mechanism must be proven to be valid in adult or old mice to validate its involvement in the pathogenesis of anxiety/AD.

# Methods

**Key resources table**

| Reagent type (species) or resource | Designation | Source or reference | Identifiers | Additional information |
|---|---|---|---|---|
| Strain, strain background (*Mus musculus*) | C57BL6 | Japan SLC | RRID:IMSR_JAX:000664 | Female and male |
| Antibody | Mouse anti-TH (monoclonal) | Santa Cruz Biotechnology | Cat# sc-25269, RRID:AB_628422 | IHC (1:500), |
| Antibody | Rabbit anti-sodium potassium ATPase (monoclonal) | Abcam | Cat# ab76020, RRID:AB_1310695 | IHC (1:1000), WB (1:1000) |
| Antibody | Goat anti-α2A AR (monoclonal) | Abcam | Cat# ab45871, RRID:AB_722745 | IHC (1:200) |
| Antibody | Rabbit anti-α2A AR (monoclonal) | Neuromics | Cat# RA14110, RRID:AB_2225052 | IHC (1:500) |
| Antibody | Cy5 donkey anti-mouse IgG (polyclonal) | Vector Laboratories | Cat# cy-2500, RRID:AB_11099905 | IHC (1:500) |
| Antibody | FITC donkey anti-rabbit IgG (polyclonal) | Jackson ImmunoResearch | Cat# 711-095-152, RRID:AB_2315776 | IHC (1:500) |
| Antibody | Cy3 donkey anti-goat IgG (polyclonal) | Jackson ImmunoResearch | Cat# 711-095-152, RRID:AB_2307351 | IHC (1:500) |
| Antibody | Rabbit anti-MAO-A (monoclonal) | Abcam | Cat# ab126751, RRID:AB_11129867 | IHC (1:2000) |
| Antibody | FITC donkey anti-mouse IgG (polyclonal) | Jackson ImmunoResearch | Cat# 715-545-150, RRID:AB_2340846 | IHC (1:500) |
| Antibody | Cy3 donkey anti-rabbit IgG (polyclonal) | Jackson ImmunoResearch | Cat# 711-165-152, RRID:AB_2307443 | IHC (1:2000) |
| Antibody | Mouse anti-legumain (monoclonal) | Santa Cruz Biotechnology | Cat# sc-133234, RRID:AB_213501 | IHC (1:200) |
| Antibody | Rabbit anti-DBH (polyclonal) | ImmunoStar | Cat# 22806, RRID:AB_572229 | IHC (1:500) |
| Antibody | Anti-rabbit IgG, HRP-linked (polyclonal) | Cell Signaling Technology | Cat# 7074, RRID:AB_2099233 | WB (1:5000) |
| Antibody | Rabbit anti-α2A AR (polyclonal) | alomone | Cat# AAR-020, RRID:AB_10687546 | WB (1:500) |
| Antibody | Rabbit anti-GAPDH (polyclonal) | Abfrontier | Cat# LF-PA0018, RRID:AB_161673 | WB (1:1000) |
| Antibody | Mouse anti-beta actin (monoclonal) | MilliporeSigma | Cat# A5441, RRID:AB_476744 | WB (1:2000) |

*Continued on next page*

*Continued*

| Reagent type (species) or resource | Designation | Source or reference | Identifiers | Additional information |
|---|---|---|---|---|
| Antibody | Rabbit anti-TH (polyclonal) | GeneTex | Cat# GTX113016 RRID:AB_1952230 | WB (1:1600) |
| Antibody | Rabbit anti-MAO-A (monoclonal) | Abcam | Cat# ab126751, RRID:AB_11129867 | WB (1:6400) |
| Antibody | Mouse anti-legumain (monoclonal) | Santa Cruz | Cat# sc-133234, RRID:AB_213501 | WB (1:400) |
| Antibody | Mouse anti-TAU-5 (monoclonal) | Santa Cruz Biotechnology | Cat# SC-58860, RRID:AB_785931 | WB (1:1500) |
| Antibody | Mouse anti-beta actin (monoclonal) | Proteintec, Rosemont | Cat# 66009–1-Ig, RRID:AB_2687938 | WB (1:1280) |
| Antibody | Rabbit anti-tau AEP-cleaved (N368) (monoclonal) | MilliporeSigma | Cat# ABN1703 | WB (1:5000) |
| Chemical Compound, Drug | Atipamezole | MilliporeSigma | A9611 | Electrophysiology |
| Chemical Compound, Drug | Barbadin | MilliporeSigma | SML3127 | Electrophysiology |
| Chemical Compound, Drug | Tertiapin-Q | Tocris Bioscience | 1316 | Electrophysiology |
| Software; Algorithm | Axograph X | Axograph | RRID:SCR_014284 | Electrophysiology |

## Restraint stress

Three-week-old male C57BL/6 mice (JAPAN HAMAMATSU SLC, Hamamatsu, Japan) were used and housed at 24 ± 2°C. Food and water were available ad libitum. The mice in the stressed groups were restrained in a modified 50 ml clear polystyrene conical centrifuge with multiple air holes for ventilation. The mice were subjected to 12 hr RS (20:00 p.m. to 8:00 a.m. of the next day) for 1, 2, 3, and 5 days. In the previous study, the duration of RS widely ranged from 0.5 (*Gray et al., 2010*) to 24 hr (*Chu et al., 2016*), while the RS with the duration less than 12 hr has been applied daily for a period: 6 hr for 2 weeks (*Sakaguchi and Nakamura, 1990*) 8 hr for up to 16 weeks (*Sugama et al., 2016*) 12 hr for 2 days (*Zhang et al., 2008*). As the prolonged stress is known to be a potential risk factor for AD (*Bisht et al., 2018*), the protocol of daily 12 hr RS is relevant for the study of functional and cytological disorders of LC neurons. Once the restraint ended, the mice were returned to their home cages with access to food and water ad libitum. The control group remained in their home cages without the restraint procedure.

## Slice preparation and electrophysiological recordings

Slices were prepared from 3-week-old male C57BL/6 mice to investigate the effects of RS for varying periods on GIRK-I and from C57BL/6 mice of either sex at 16–23 postnatal days to investigate the effects of $[Ca^{2+}]_i$ increases on GIRK-I, similar to our previous study (*Toyoda et al., 2022*). Using an Axopatch 200B (MDS Analytical Technologies, Sunnyvale, CA, USA), whole-cell recordings were made from LC neurons that were viewed under Nomarski optics (BX51WI; Olympus, Tokyo, Japan). LC neurons were identified immunohistochemically and electrophysiologically as we previously reported (*Toyoda et al., 2022*). The internal solution of the patch pipettes had the following ionic composition (in mM): 123 K-gluconate, 18 KCl, 10 NaCl, 2 $MgCl_2$, 2 ATP-$Na_2$, 0.3 GTP-$Na_3$, 10 HEPES, and 0.2 EGTA; pH 7.3. The membrane potential values given in the text were corrected for the junction potential (10 mV) between the internal solution for the whole-cell recording (negative) and the standard extracellular solution. The pipette resistances were 4–6 MΩ. The series resistance was <10 MΩ. All recordings were made at room temperature (RT). Records of currents and voltages were low-pass filtered at 5 kHz (3-pole Bessel filter), digitized at a sampling rate of 10 kHz (Digidata 1322A, MDS Analytical Technologies), and stored on a computer hard disk. Spike-frequency adaptation was quantitatively estimated by fitting with a saturation curve defined by Monod equation ($y = (a * x)/(b+x)$) with a saturation level ($a$) and a half saturation constant ($b$). The amplitudes of inward and outward components of NA-induced GIRK-I were measured at –130 and –60 mV.

## Procedure for immunohistochemical examination

C57BL6/J male mice (4–5 weeks) were perfused transcardially with 5 ml of PBS, followed by 10 ml of a freshly prepared 4% paraformaldehyde in PBS. The whole brain, including LC (–9 to –10 mm from bregma), was dissected out and immersed in 4% paraformaldehyde solution for 4 hr at 4°C followed by cryoprotection with 30% (wt/wt) sucrose in PBS.

Brain stem, including LC regions, was cut into 8- to 20-μm-thick coronal sections on a freezing microtome (Leica Microsystems, Wetzlar, Germany). To examine the distribution of α2A-AR in LC neurons, mouse anti-TH (1:500; sc-25269, Santa Cruz Biotechnology, Santa Cruz, CA, USA), rabbit anti-sodium potassium ATPase (1:1000; ab76020, Abcam, Cambridge, MA, USA), and goat anti-α2A-AR (1:200; ab45871, Abcam) or rabbit anti-α2A-AR (1:500; RA14110, Neuromics, Northfield, MN, USA) were used as primary antibodies, while cy5 donkey anti-mouse IgG (1:500; cy-2500, Vector Laboratories, Burlingame, CA, USA), FITC donkey anti-rabbit IgG (1:500; 711-095-152, Jackson ImmunoResearch, West Grove, PA, USA), and Cy3 donkey anti-goat IgG (1:500; 706-165-147, Jackson ImmunoResearch) were used as secondary antibodies. We used the affinity-purified IgG antibody for α2A-AR immunohistochemistry, in which antibody specificity was ensured as examined previously (*Stone et al., 1998*; *Tan et al., 2009*). To examine the expression in LC neurons, mouse anti-TH (1:500) and rabbit anti-MAO-A (1:2000; ab126751, Abcam) were used as primary antibodies, while FITC donkey anti-mouse IgG (1:500; 715-545-150; Jackson ImmunoResearch) for TH and Cy3 donkey anti-rabbit IgG (1:2000; 711-165-152; Jackson ImmunoResearch) were used as secondary antibodies. To examine the distribution of legumain in LC neurons, mouse anti-legumain (1:200; sc-133234, Santa Cruz) and rabbit anti-DBH (1:500; 22806, ImmunoStar, Hudson, WI, USA) were used as primary antibodies, while FITC donkey anti-mouse IgG (1:500; 715-545-150, Jackson ImmunoResearch) and Cy3 donkey anti-rabbit IgG (1:500; 711-165-152, Jackson ImmunoResearch) were used as secondary antibodies. The slices were coverslipped with mounting media (H1000, Vector Laboratories), and sections were visualized with a confocal microscope (LSM980, Carl Zeiss, Oberkochen, Germany) at ×10 and ×63 magnification.

## Western blot study

Brain slices from the control and 5-day RS-treated mice were sectioned with vibratome (VT1000P, Leica Microsystems) to obtain 500-μm-thick coronal sections containing LC region (bregma –5.34 mm to –5.80 mm). We performed a western blot study only in 5-day RS mice without examining in 3-day RS mice, under the assumption that for the degeneration-related biochemical changes to reach sufficiently detectable levels, a high $[Ca^{2+}]_i$ condition due to the rundown of GIRK-I has to be sustained for some period of time after the rundown of GIRK-I reached a plateau level in 3-day RS mice (see *Figure 4F and G*). LC dissection was done via micro punch using 1.5 mm biopsy punch (BP-15F, Kai Medical, Gifu, Japan) bilaterally under light microscope. Dissected LC was quickly frozen in liquid nitrogen and stored at −80°C until processed. Because the LC region was too small to collect sufficient tissue from one mouse for the analysis, the punched LC tissues were collected from two to three mice to obtain a protein lysate sample for each western blotting analysis. LC tissues were homogenized by bead homogenizer in T-PER Tissue Protein Extraction Reagent (78510, Thermo Fisher) containing protease and phosphatase inhibitor cocktails (MilliporeSigma, St. Louis, MO, USA). The membrane and cytosol proteins from LC lysate were isolated using Mem-PER Plus Kit (78840, Thermo Fisher) in accordance with the manufacturer's protocol. The protein concentration of the lysate was determined using a Lowry protein assay (Bio-Rad, Hercules, CA, USA). Protein samples (20 μg) were heated to 95°C for 5 min, separated by 12% sodium dodecyl sulfate-polyacrylamide gel electrophoresis, and transferred to a polyvinylidene fluoride membrane using a Transblot SD apparatus (Bio-Rad). After the blots had been washed with Tris-buffered saline containing Tween-20 (TBST, 10 mM Tris-HCl, 150 mM NaCl, 0.05% Tween-20), the membranes were blocked with 2% BSA and 4% skim milk/TBST at RT for 1 hr and incubated at 4°C overnight with primary antibodies, followed by an anti-rabbit IgG horseradish peroxidase (HRP)-conjugated secondary antibody (1:5,000, 7074; Cell Signaling Technology, Danvers, MA, USA) at RT for 1 hr. The western blot membrane was stripped of the primary and secondary antibodies and reprobed with the anti-β-actin antibody followed by the anti-rabbit IgG HRP-conjugated secondary antibody. The following primary antibodies were used for the blots: rabbit anti-α2A-AR (1:500; AAR-020, alomone), rabbit anti-GAPDH (1:1000; LF-PA0018, Abfrontier, Seoul, Korea), mouse anti-beta actin (1:2000, A5441, MilliporeSigma), and rabbit anti-sodium potassium ATPase (1:1000; ab76020, Abchem) (*Figure 3C, F, and G*); rabbit anti-TH (1:1600; GTX113016, GeneTex, Irvine, CA,

USA), rabbit anti-MAO-A (1:6400; ab126751, Abcam), mouse anti-legumain (AEP) (1:400; sc-133234, Santa Cruz), mouse anti-TAU-5 (1:1500, Santa Cruz Biotechnology, SC-58860), and mouse anti-beta actin (1:12800, 66009-1-Ig, Proteintec, Rosemont, IL, USA) (*Figure 5A, D and G*); rabbit anti-tau AEP-cleaved (N368) (1:5000, ABN1703, MilliporeSigma) and mouse anti-beta actin (1:1000, A5441, MilliporeSigma) (*Figure 5G and I*). Western blot analysis was performed by measuring band intensity with ImageJ 1.50i software (National Institutes of Health, Bethesda, MD, USA).

## Quantitative real-time PCR

cDNA was synthesized with Super Script II Reverse Transcriptase (18064, Invitrogen, Carlsbad, CA, USA) with 1 mg of LC neuron RNA. Using a 7300 Gene Amp PCR system (Applied Biosystems, Carlsbad), quantitative real-time PCR was run in Power SYBR Green PCR master mix (4367656; Life Technologies, Carlsbad) under the following thermal cycling conditions; an initial step of 94°C/5 min, followed by 40 cycles of a series of steps of 94°C/45 s, 61°C/45 s, 72°C/45 s and ended with a series of steps of 95°C/60 s, 60°C/60 s, 95°C/15 s, 60°C/15 s. The following primers were used: α2A (forward GGAATCATGGCTGTGGAGAT, reverse CAGAAATCCCTTCCCTGTCA); α2C (forward TCAT CGTTTTCACCGTGGTA, reverse GCTCATTGGCCAGAGAAAAG); GIRK1 (forward ACCCTGGTGGAT CTCAAGTG, reverse GGCCACACAGGGAGTGTAGT); GRIK2 (forward CAACCTCAACGGGTTTGTCT , reverse ATCCTACCATGAAGGCGTTG); GIRK3 (forward CGTCTCACCTCTCGTCATCA, reverse CATCCACCAGGTACGAGCTT); GAPDH (forward CCAGAACATCATCCCTGCAT, reverse GCATCGAA GGTGGAAGAGTG).

## Behavioral test

The control mice as non-stress group and mice exposed to RS for 1 day, 3 days, and 5 days as 1-day, 3-day, and 5-day RS groups were tested on the Y maze apparatus, consisting of three identical arms in a 'Y' shape made of white acryl (35 cm in length, 5 cm in width, and 20 cm in height) to assess working spatial memory. All tests were performed between 9 a.m. and noon. Investigators were blinded to the group when performing experiments. Each mouse was placed at the extremity of the 'start arm' and left for a training session of 15 min, with one of the two other arms closed with a white acrylic board (the closed arm was randomly chosen for each mouse). Then, the mouse was brought back to its cage for 5 min, the maze was cleaned, the closed arm was opened, and the mouse was placed at the extremity of the start arm for a 5 min test session, under video tracking. The time spent and distance traveled in each arm (start, familiar, and novel arms) were measured and analyzed with Smart 3.0 video tracking software (Panlab, Harvard Apparatus, Holliston, MA, USA). Given their natural curiosity, mice are expected to spend more time exploring the new arm.

## Drug application

Atipamezole as a blocker of α2-AR and tertiapin-Q as a GIRK blocker were bath-applied at 5–10 µM and 200 nM, respectively. Barbadin as an arrestin/AP2 blocker was added to the internal solution at a final concentration of 100 µM (*Beautrait et al., 2017*). All chemicals except tertiapin-Q were purchased from MilliporeSigma unless specified otherwise. Tertiapin-Q was purchased from Tocris Bioscience (Bristol, UK).

## Statistical analysis

Statistical analysis was performed using STATISTICA10J (StatSoft Japan, Tokyo, Japan). Numerical data were expressed as the mean ± SD. The statistical significance was assessed using Student's t-test, one-way ANOVA followed by post hoc Fisher's PLSD and Wilk's lambda. The significance level was set at 5% or less ($p < 0.05$). Statistical results are given as a p-value, unless $p < 0.001$.

## Acknowledgements

This research was funded by Japan Society for the Promotion of Science (26290006 and 21K06441 to YK: 20K06926 and 23K06346 to HT). YK was supported by the Brain Pool Program (NRF) funded by the Korean government (Ministry of Science and ICT). HT was supported by the Naito Grant funded by the Naito Foundation. This research was partly supported by the National Research Foundation of Korea (KRF) (RS-2023-00264409 and RS-2023-00302281 to SBO) funded by the Korean government (Ministry of Science and ICT).

# Additional information

## Funding

| Funder | Grant reference number | Author |
|---|---|---|
| Japan Society for the Promotion of Science | 20K06926 | Hiroki Toyoda |
| Japan Society for the Promotion of Science | 23K06346 | Hiroki Toyoda |
| Japan Society for the Promotion of Science | 26290006 | Youngnam Kang |
| Japan Society for the Promotion of Science | 21K06441 | Youngnam Kang |
| Ministry of Science and ICT | Brain Pool Program | Youngnam Kang |
| Naito Foundation | Naito Grant | Hiroki Toyoda |
| Ministry of Science and ICT | RS-2023-00264409 | Seog Bae Oh |
| Ministry of Science and ICT | RS-2023-00302281 | Seog Bae Oh |

The funders had no role in study design, data collection and interpretation, or the decision to submit the work for publication.

## Author contributions

Hiroki Toyoda, Conceptualization, Data curation, Formal analysis, Funding acquisition, Validation, Writing – original draft, Writing – review and editing; Doyun Kim, Byeong Geon Koh, Data curation, Formal analysis, Writing – review and editing; Tomomi Sano, Takashi Kanematsu, Data curation, Formal analysis, Writing – original draft, Writing – review and editing; Seog Bae Oh, Conceptualization, Supervision, Funding acquisition, Writing – original draft, Writing – review and editing; Youngnam Kang, Conceptualization, Formal analysis, Supervision, Funding acquisition, Validation, Writing – original draft, Writing – review and editing

## Author ORCIDs

Hiroki Toyoda https://orcid.org/0000-0002-9907-7176
Byeong Geon Koh https://orcid.org/0000-0002-8801-660X
Seog Bae Oh https://orcid.org/0000-0001-7975-6895
Youngnam Kang https://orcid.org/0000-0002-6994-6713

## Ethics

This study was conducted following the Helsinki Declaration, and carried out according to the ARRIVE guidelines. All experimental procedures were approved and performed in accordance with the relevant guidelines of the Ethical Guidelines Committee at Osaka University (project identification code: H27-012-0) and Seoul National University (project identification code: SNU-181231-1-3). Every possible effort was made to reduce the suffering of the animals.

Reviewer #1 (Public review): https://doi.org/10.7554/eLife.106362.4.sa1
Reviewer #2 (Public review): https://doi.org/10.7554/eLife.106362.4.sa2
Reviewer #3 (Public review): https://doi.org/10.7554/eLife.106362.4.sa3
Author response https://doi.org/10.7554/eLife.106362.4.sa4

# Additional files

## Supplementary files

MDAR checklist

## Data availability

All data generated or analyzed during this study are included in the manuscript and supporting files. Original western blot images are uploaded as zipped source data files.

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
