## [Editor Report · eLife Assessment]

This **fundamental** study explores a novel cellular mechanism underlying the degeneration of locus coeruleus neurons during chronic restraint stress. The evidence supporting the overexcitation of LC neurons after chronic stress is **compelling**. The topic is timely, the proposed mechanistic pathway is innovative, and the findings have translational relevance, particularly regarding therapeutic strategies targeting α2A-AR internalization in neurodegenerative diseases.

---

## [Referee Report · Reviewer #1 (Public review)]

This study aims to elucidate the mechanisms by which stress-induced α2A-adrenergic receptor (α2A-AR) internalization leads to cytosolic noradrenaline (NA) accumulation and subsequent neuronal dysfunction in the locus coeruleus (LC). While the manuscript presents an interesting but ambitious model involving calcium dynamics, GIRK channel rundown, and autocrine NA signaling, several key limitations undermine the strength of the conclusions.

First, the revision does not include new experiments requested by reviewers to validate core aspects of the mechanism. Specifically, there is no direct measurement of cytosolic NA levels or MAO-A enzymatic activity to support the link between receptor internalization and neurochemical changes. The authors argue that such measurements are either not feasible or beyond the scope of the study, leaving a significant gap in the mechanistic chain of evidence.

Second, the behavioral analysis remains insufficient to support claims of cognitive impairment. The use of a single working memory test following an anxiety test is inadequate to verify memory dysfunction behaviors. Additional cognitive assays, such as the Morris Water Maze or Novel Object Recognition, are recommended but not performed.

Third, concerns regarding the lack of rigor in differential MAO-A expression in fluorescence imaging were not addressed experimentally. Instead of clarifying the issue, the authors moved the figure to supplementary data without providing further evidence (e.g., an enzymatic assay or quantitative reanalysis of Western blot, or re-staining of IF for MAO-A) to support their interpretation.

Fourth, concerns regarding TH staining remain unresolved. In Figure S7, the α2A-AR signal appears to resemble TH staining, and vice versa, raising the possibility of labeling errors. It is recommended that the authors re-examine this issue by either double-checking the raw data or repeating the immunostaining to validate the staining.

Overall, the manuscript offers a potentially interesting framework but falls short in providing the experimental rigor necessary to establish causality. The reliance on indirect reasoning and reorganizing existing data, rather than generating new evidence, limits the overall impact and interpretability of the study.

---

## [Referee Report · Reviewer #2 (Public review)]

Summary:

This manuscript investigates the mechanism by which chronic stress induces degeneration of locus coeruleus (LC) neurons. The authors demonstrate that chronic stress leads to the internalization of α2A-adrenergic receptors (α2A-ARs) on LC neurons, causing increased cytosolic noradrenaline (NA) accumulation and subsequent production of the neurotoxic metabolite DOPEGAL via monoamine oxidase A (MAO-A). The study suggests a mechanistic link between stress-induced α2A-AR internalization, disrupted autoinhibition, elevated NA metabolism, activation of asparagine endopeptidase (AEP), and Tau pathology relevant to Alzheimer's disease (AD). The conclusions of this paper are well-supported mainly by the data, but some aspects of image acquisition require further examination.

Strengths:

This study clearly demonstrates the effects of chronic stimulation on the excitability of LC neurons using electrophysiological techniques. It also elucidates the role of α2-adrenergic receptor (α2-AR) internalization and the associated upstream and downstream signaling pathways of GIRK-1, using a range of pharmacological agents, highlighting the innovative nature of the work. Additionally, the study identifies the involvement of the MAO-A-DOPEGAL-AEP pathway in this process. The topic is timely, the proposed mechanistic pathway is compelling, and the findings have translational relevance, particularly in relation to therapeutic strategies targeting α2A-AR internalization in neurodegenerative diseases.

Weaknesses:

(1) The manuscript reports that chronic stress for 5 days increases MAO-A levels in LC neurons, leading to the production of DOPEGAL, activation of AEP, and subsequent tau cleavage into the tau N368 fragment, ultimately contributing to neuronal damage. However, the authors used wild-type C57BL/6 mice, and previous literature has indicated that AEP-mediated tau cleavage in wild-type mice is minimal and generally insufficient to cause significant behavioral alterations. Please clarify and discuss this apparent discrepancy.

(2) It is recommended that the authors include additional experiments to examine the effects of different durations and intensities of stress on MAO-A expression and AEP activity. This would strengthen the understanding of stress-induced biochemical changes and their thresholds.

(3) Please clarify the rationale for the inconsistent stress durations used across Figures 3, 4, and 5. In some cases, a 3-day stress protocol is used, while in others, a 5-day protocol is applied. This discrepancy should be addressed to ensure clarity and experimental consistency.

(4) The abbreviation "vMAT2" is incorrectly formatted. It should be "VMAT2," and the full name (vesicular monoamine transporter 2) should be provided at first mention.

Comments on revisions:

The authors have addressed all of the reviewers' comments.

---

## [Referee Report · Reviewer #3 (Public review)]

Summary:

The authors present a technically impressive dataset showing that repeated excitation or restraint stress internalises somatodendritic α2A adrenergic autoreceptors (α2A ARs) in locus coeruleus (LC) neurons. Loss of these receptors weakens GIRK-dependent autoinhibition, raises neuronal excitability, and is accompanied by higher MAO A, DOPEGAL, AEP, and tau N368 levels. The work combines rigorous whole-cell electrophysiology with barbadin-based trafficking assays, qPCR, Western blotting, and immunohistochemistry. The final schematic is appealing and, in principle, could explain early LC hyperactivity followed by degeneration in ageing and Alzheimer's disease.

Strengths:

- Multi-level approach - The study integrates electrophysiology, pharmacology, mRNA quantification, and protein-level analysis.

-Use of barbadin to block β-arrestin/AP-2-dependent internalisation is both technically precise and mechanistically informative

-Well-executed electrophysiology

-translation relevance

-converges to a model that peers discussed (scientists can only discuss models - not data!)

Weaknesses:

Nevertheless, the manuscript currently reads as a sequence of discrete experiments rather than a single causal chain.

---

## [Author Response]

The following is the authors’ response to the previous reviews

**Reviewer # 1 (Public review)**
This study aims to elucidate the mechanisms by which stress-induced α2A-adrenergic receptor (α2A-AR) internalization leads to cytosolic noradrenaline (NA) accumulation and subsequent neuronal dysfunction in the locus coeruleus (LC). While the manuscript presents an interesting but ambitious model involving calcium dynamics, GIRK channel rundown, and autocrine NA signaling, several key limitations undermine the strength of the conclusions.(1) First, the revision does not include new experiments requested by reviewers to validate core aspects of the mechanism. Specifically, there is no direct measurement of cytosolic NA levels or MAO-A enzymatic activity to support the link between receptor internalization and neurochemical changes. The authors argue that such measurements are either not feasible or beyond the scope of the study, leaving a significant gap in the mechanistic chain of evidence.

Although the reviewer #1 commented that “The authors argue that such measurements are either not feasible or beyond the scope of the study, leaving a significant gap in the mechanistic chain of evidence”, we believe that this comment may be unfair.

It may be unfair for the reviewer #1 to neglect our responses to the original reviewer comments regarding the direct measurement of cytosolic NA levels. It is true that none of the recommended methods to directly measure cytosolic NA levels are not feasible as described in the original authors’ response (see the original authors’ response to the comment raised by the Reviewer #1 as Recommendations for the authors (2)). To measure extracellular NA with GRAB-NE photometry, α2A-ARs must be expressed in the cell membrane. GRAB-NE photometry is not applicable unless α2A-ARs are expressed, whereas increases in cytosolic NA levels are caused by internalization of α2A-ARs in our study.

In our study, we elaborated to detect the change in MAO-A protein with Western blot method, instead of examining MAO-A enzymatic activity. Because the relative quantification of active AEP and Tau N368 proteins by Western blot analysis should accurately reflect the change in the MAO-A enzymatic activity, enzymatic assay may not be necessarily required while we admit the necessity of enzymatic assay to better demonstrate the MAO-A activities as discussed in the previously revised manuscript (R1, page 10, lines 314-315).

We used the phrase “beyond the scope of the current study” for “the mechanism how Ca^2+^ activates MAO-A” as described in the original authors’ responses (see the original authors’ response to the comment raised by the Reviewer #1 as Weakness (3)). We do not think that this mechanism must be investigated in the present study because the Ca^2+^ dependent nature of MAO-A activity is already known (Cao et al., 2007).

On the other hand, because it is not possible to measure cytosolic NA levels with currently available methods, the quantification of the connection between α2A-AR internalization and increased cytosolic NA levels must be considered outside the scope of the study. However, our study demonstrated the qualitative relationship between α2A-AR internalization and active-AEP/TauN-368 reflecting increased cytosolic NA levels, leaving “a small gap in the mechanistic chain of evidence.” Therefore, it may be unreasonable to criticize our study as “leaving a significant gap in the mechanistic chain of evidence” with the phrase “beyond the scope of the current study.”

(2) Second, the behavioral analysis remains insufficient to support claims of cognitive impairment. The use of a single working memory test following an anxiety test is inadequate to verify memory dysfunction behaviors. Additional cognitive assays, such as the Morris Water Maze or Novel Object Recognition, are recommended but not performed.

As described in the original authors’ response (see the original authors’ response to the comment raised by the Reviewer #1 as Weakness (4)), we had already done another behavioral test using elevated plus maze (EPM) test. By combining the two tests, it may be possible to more accurately evaluate the results of Y-maze test by differentiating the memory impairment from anxiety. However, the results obtained by these behavioral tests showed that chronic RS mice displayed both anxiety-like and memory impairment-like behaviors. Accordingly, we have softened the implication of anxiety and memory impairment (page 13, lines 396-399) and revised the abstract (page 2, line 59) in the revised manuscript (R2).

(3) Third, concerns regarding the lack of rigor in differential MAO-A expression in fluorescence imaging were not addressed experimentally. Instead of clarifying the issue, the authors moved the figure to supplementary data without providing further evidence (e.g., an enzymatic assay or quantitative reanalysis of Western blot, or re-staining of IF for MAO-A) to support their interpretation.

Because the quantification of MAO-A expression can be performed with greater accuracy by means of Western blot than by immunohistochemistry, we have moved the immunohistochemical results (shown in Figure 5) to the supplemental data (Figure S8) following the suggestion made by the Reviewer #3. As the relative quantification of active AEP and Tau N368 proteins by Western blot analysis may accurately reflect changes in the MAO-A enzymatic activity which is consistent with the result of Western blot analysis of MAO-A, enzymatic assay or re-staining of immunofluorescence for MAO-A may not be necessarily required. We do not think that a new experiment of Western blot analysis is necessary to re-evaluate MAO-A just because of the lack of the less-reliable quantification of immunohistochemical staining.

(4) Fourth, concerns regarding TH staining remain unresolved. In Figure S7, the α2A-AR signal appears to resemble TH staining, and vice versa, raising the possibility of labeling errors. It is recommended that the authors re-examine this issue by either double-checking the raw data or repeating the immunostaining to validate the staining.

The reviewer #3 is misunderstanding Figure S7. In Figure S7, there are two types of α2A-AR expressing neurons; one is TH-positive LC neuron and the other is TH-negative neuron in mesencephalic trigeminal nucleus (MTN). This clearly indicates that TH staining is specific. Furthermore, α2A-AR staining was much more extensive in MTN neurons than in LC neurons. Thus, α2A-AR signal is not similar to TH signal and there are no labeling errors, which is also evident in the merged image (Figure S7C).

(5) Overall, the manuscript offers a potentially interesting framework but falls short in providing the experimental rigor necessary to establish causality. The reliance on indirect reasoning and reorganizing of existing data, rather than generating new evidence, limits the overall impact and interpretability of the study.

Overall, the reviewer #1 was not satisfied with our revision regardless of the authors’ responses. As detailed above in our responses to the replies (1)~(4), we believe that in the original authors’ responses and in the above-described responses we effectively responded to the criticisms by the reviewer #1.

**Reviewer #2 (Public review):**
Comments on revisions:The authors have addressed all of the reviewers' comments.

We appreciate constructive and helpful comments made by the reviewer #2.

**Reviewer #3 (Public review):**
Weaknesses:Nevertheless, the manuscript currently reads as a sequence of discrete experiments rather than a single causal chain. Below, I outline the key points that should be addressed to make the model convincing.

Please see the responses to the recommendation for the authors made by reviewer #3.

**Reviewer #3 (Recommendations for the authors):**
(1) Causality across the pathwayEach step (α2A internalisation, GIRK rundown, Ca^2+^ rise, MAO-A/AEP upregulation) is demonstrated separately, but no experiment links them in a single preparation. Consider in vivo Ca^2+^ or GRAB NE photometry during restraint stress while probing α2A levels with i.p. clonidine injection or optogenetic over excitation coupled to biochemical readouts. Such integrated evidence would help to overcome the correlational nature of the manuscript to a more mechanistic study.Authors response: It is not possible to measure free cytosolic NA levels with GRAB NE photometry when α2A AR is internalized as described above (see the response to the comment made by reviewer #1 as the recommendation for the authors).The core idea behind my comment, as well as that of Reviewer 1, was to encourage integrating your individual findings into a more cohesive in vivo experiment. Using GRAB-NE to measure extracellular NA could serve as an indirect readout of NA uptake via NAT, and ultimately, cytosolic NA levels. Connecting these experiments would significantly strengthen the manuscript and enhance its overall impact.

It may be true that the measurement of extracellular NA could serve as an indirect readout of NA uptake via NAT, and ultimately cytosolic NA levels. However, the reviewer #3 is still misunderstanding the applicability of GRAB-NE method to detect NE in our study. As described in the original authors’ response, there appeared to be no fluorescence probe to label cytosolic NA at present. Especially, the GRAB-NE method recommended by the reviewers #1 and #3 is limited to detect NA only when α2A-AR is expressed in the cell membrane.Therefore, when increases in cytosolic NA levels are caused by internalization of α2A-ARs, NA measurement with GRAB-NE photometry is not applicable.

(2) Pharmacology and NE concentrationThe use of 100 µM noradrenaline saturates α and β adrenergic receptors alike. Please provide ramp measurements of GIRK current in dose-response at 1-10 µM NE (blocked by atipamezole) to confirm that the rundown really reflects α2A activity rather than mixed receptor effects.Authors response: It is true that 100 µM noradrenaline activates both α and β adrenergic receptors alike. However, it was clearly showed that enhancement of GIRK-I by 100 µM noradrenaline was completely antagonized by 10 µM atipamezole and the Ca^2+^ dependent rundown of NA-induced GIRK-I was prevented by 10 µM atipamezole. Considering the Ki values of atipamezole for α2A AR (=1~3 nM) (Vacher et al., 2010, J Med Chem) and β AR (>10 µM) (Virtanen et al., 1989, Arch Int Pharmacodyn Ther), these results really reflect α2A AR activity but not β AR activity (Figure S5). Furthermore, because it is already well established that NA-induced GIRK-I was mediated by α2A AR activity in LC neurons (Arima et al., 1998, J Physiol; Williams et al., 1985, Neuroscience), it is not necessarily need to re-examine 1-10 µM NA on GIRK-I.While the milestone papers by Williams remain highly influential, they should be re-evaluated in light of more recent findings, given that they date back over 40 years. Advances in our understanding now allow for a more nuanced interpretation of some of their results. For example, see McKinney et al. (eLife, 2023). This study demonstrates that presynaptic β-adrenergic receptors-particularly β2-can enhance neuronal excitability via autocrine mechanisms. This suggests that your post-activation experiments using atipamezole may not fully exclude a contribution of β-adrenergic signaling. Such a role might become apparent when conducting more detailed titration experiments.

The reviewer #3 may be misunderstanding the report by McKinney et al. (eLife, 2013). This paper did not demonstrate that presynaptic β-adrenergic receptors-particularly β2- can enhance neuronal excitability via autocrine mechanisms. It is impossible for LC neurons to increase their excitability by activating β-adrenergic receptors, as we have clearly shown that enhancement of GIRK-I by 100 µM noradrenaline was completely antagonized by 10 µM atipamezole. Considering the difference in Ki values of atipamezole for α2-AR (= 2~4 nM) (Vacher et al., 2010, J Med Chem) and β-AR (>10 µM) (Virtanen et al., 1989, Arch Int Pharmacodyn Ther), such a complete antagonization (of 100 µM NA-induced GIRK-I) by 10 µM atipamezole really reflect α2A-AR activity but not β-AR activity (Figure S5). Furthermore, it is already well established that NA-induced GIRK-I was mediated by α2-AR activity in LC neurons (Arima et al., 1998, J Physiol). McKinney et al. (eLife, 2023) have just found the absence of lateral inhibition on adjacent LC neurons by NA autocrine caused respective spike activity. This has nothing to do with autoinhibition.

(4) Age mismatch and disease claimsAll electrophysiology and biochemical data come from juvenile (< P30) mice, yet the conclusions stress Alzheimer-related degeneration. Key endpoints need to be replicated in adult or aged mice, or the manuscript should soften its neurodegenerative scope.Authors response: As described in the section of Conclusion, we never stress Alzheimer-related degeneration, but might give such an impression. To avoid such a misunderstanding, we have added a description “However, the present mechanism must be proven to be valid in adult or old mice, to validate its involvement in the pathogenesis of AD.” (R1, page 14, lines 448-450).It would be great to see this experiment performed in aged mice-you are the one who has everything in place to do it right now!

In our future separate studies, we would like to prove that the present mechanism is valid in aged mice, to validate its involvement in the pathogenesis of AD. This is partly because the patch-clamp study in aged mice is extremely difficult and takes much time.

Authors response: In the abstract, you suggest that internalization of α2A-adrenergic receptors could represent a therapeutic target for Alzheimer's disease. "...Thus, it is likely that internalization of α2A-AR increased cytosolic NA, as reflected in AEP increases, by facilitating reuptake of autocrine-released NA. The suppression of α2A-AR internalization may have a translational potential for AD treatment."

α2A-AR internalization was involved in the degeneration of LC neurons. Because we confirmed that spike-frequency adaptation reflecting α2A-AR-mediated autoinhibition can be induced in adult mice as prominently as in juvenile mice (Figure S10), it is not inadequate to suggest that the suppression of α2A-AR internalization may have a translational potential for anxiety/AD treatment (see Discussion; R2, page 14, lines 445-449).

(6) Quantitative histologyFigure 5 presents attractive images, but no numerical analysis is provided. Please provide ROI-based fluorescence quantification (with n values) or move the images to the supplement and rely on the Western blots.Author response: We have moved the immunohistochemical results in Fig. 5 to the supplement, as we believe the quantification of immunohistochemical staining is not necessarily correct.What do you mean by that " ...immunohistochemical staining is not necessarily correct."

It is evident that in terms of quantification, Western blot analysis is a more accurate method than immunohistochemical staining. In this sense, it is the contention of our study that the ROI-based fluorescence quantification of immunohistochemical staining is not necessarily an accurate or correct procedure, compared to the quantification by Western blot analysis.